# Functional diversification gave rise to allelic specialization in a rice NLR immune receptor pair

**Juan Carlos De la Concepcion[1,2]\*, Javier Vega Benjumea[2,3], Aleksandra Bialas[4], Ryohei Terauchi[5,6], Sophien Kamoun[4], Mark J Banfield[2]\***

[1]Gregor Mendel Institute of Molecular Plant Biology, Austrian Academy of Sciences, Vienna, Austria; [2]Department of Biological Chemistry and Metabolism, John Innes Centre, Norwich, United Kingdom; [3]Servicio de Bioquímica-Análisis clínicos, Hospital Universitario Puerta de Hierro, Madrid, Spain; [4]The Sainsbury Laboratory, University of East Anglia, Norwich, United Kingdom; [5]Division of Genomics and Breeding, Iwate Biotechnology Research Center, Iwate, Japan; [6]Laboratory of Crop Evolution, Graduate School of Agriculture, Kyoto, Japan

**Abstract** Cooperation between receptors from the nucleotide-binding, leucine-rich repeats (NLR) superfamily is important for intracellular activation of immune responses. NLRs can function in pairs that, upon pathogen recognition, trigger hypersensitive cell death and stop pathogen invasion. Natural selection drives specialization of host immune receptors towards an optimal response, whilst keeping a tight regulation of immunity in the absence of pathogens. However, the molecular basis of co-adaptation and specialization between paired NLRs remains largely unknown. Here, we describe functional specialization in alleles of the rice NLR pair Pik that confers resistance to strains of the blast fungus *Magnaporthe oryzae* harbouring AVR-Pik effectors. We revealed that matching pairs of allelic Pik NLRs mount effective immune responses, whereas mismatched pairs lead to autoimmune phenotypes, a hallmark of hybrid necrosis in both natural and domesticated plant populations. We further showed that allelic specialization is largely underpinned by a single amino acid polymorphism that determines preferential association between matching pairs of Pik NLRs. These results provide a framework for how functionally linked immune receptors undergo co-adaptation to provide an effective and regulated immune response against pathogens. Understanding the molecular constraints that shape paired NLR evolution has implications beyond plant immunity given that hybrid necrosis can drive reproductive isolation.

\*For correspondence:
juan.concepcion@gmi.oeaw.ac.
at (JCDIC);
mark.banfield@jic.ac.uk (MJB)

## Editor's evaluation

De la Concepcion and colleagues investigated the mode of co-evolution of plant immune receptor pair that functions as a unit to detect pathogen invasion and turn on immunity. The study shows that an allelic mismatch of a receptor pair from rice can cause autoimmunity in the absence of pathogen effectors, and this can be traced to polymorphisms that arose fairly recently. Overall the study supports that the paired receptors have coevolved to prevent premature inactivation.

## Introduction

Pathogens use an array of molecules, termed effectors, to successfully colonize hosts (*Win et al., 2012*). Intracellular detection of effectors relies on immune receptors from the nucleotide-binding, leucine-rich repeats (NLR) superfamily (*Bentham et al., 2020*; *Jones et al., 2016*; *Saur et al., 2021*).

Upon recognition, NLRs act as nucleotide-operated switches, exchanging ADP for ATP (***Bernoux et al., 2016***; ***Tameling et al., 2002***; ***Wang et al., 2019b***; ***Williams et al., 2011***), and oligomerize into supramolecular signalling platforms (***Hu et al., 2015***; ***Ma et al., 2020***; ***Martin et al., 2020***; ***Sharif et al., 2019***; ***Tenthorey et al., 2017***; ***Wang et al., 2019a***; ***Zhang et al., 2015***). This leads to immune responses, including programmed cell death, that restrict pathogen growth. The assembly of such sophisticated molecular machinery needs to be well coordinated and tightly regulated to ensure an efficient immune response, while avoiding the deleterious effect of constitutive immune activation (***Chae et al., 2016***; ***Karasov et al., 2017***; ***Li et al., 2020***; ***Richard and Takken, 2017***).

NLRs form the most expanded and diversified protein family in plants (***Meyers et al., 2003***; ***Van de Weyer et al., 2019***; ***Yue et al., 2012***). Since their discovery, plant NLRs have been heavily studied and around 450 NLR proteins from 31 genera of flowering plants have been functionally validated (***Kourelis et al., 2021***). Plant NLRs present multiple layers of complexity (***Barragan and Weigel, 2021***), often functioning in genetically linked pairs (***Eitas and Dangl, 2010***; ***Griebel et al., 2014***) or as part of complex immune networks (***Wu et al., 2018***). In such cases, NLRs specialize their role in immune activation, acting as 'sensors' that detect pathogen effectors or as 'helpers' that amplify and propagate immune signalling (***Adachi et al., 2019b***; ***Jubic et al., 2019***). Paired NLRs are prevalent in plant genomes (***Stein et al., 2018***; ***Wang et al., 2019c***) with a subset of sensor NLRs harbouring atypical domains integrated into their architecture (***Bailey et al., 2018***; ***Kourelis et al., 2021***; ***Kroj et al., 2016***; ***Sarris et al., 2016***). These domains can be derived from pathogen host targets that act as sensor domains within NLRs by binding pathogen effectors (***Białas et al., 2018***; ***Cesari et al., 2014***; ***Maidment et al., 2021***; ***Oikawa et al., 2020***).

Cooperating NLRs must balance a trade-off between adaptive evolution to fast-evolving pathogens and maintaining a fine-tuned regulation of complex receptor assemblies. NLRs with different evolutionary trajectories may drift apart and eventually mismatch. When these mismatched NLRs are combined in the same individual through genetic crossing, constitutive immune activation can occur, leading to deleterious phenotypes including dwarfism, necrosis, and lethality (***Bomblies et al., 2007***; ***Chae et al., 2014***). These 'Dangerous Mix' phenotypes are known in plant breeding as hybrid necrosis and have important implications in agriculture (***Caldwell and Compton, 1943***; ***Calvo-Baltanás et al., 2020***; ***Hermsen, 1963b***; ***Hermsen, 1963a***; ***Li and Weigel, 2021***; ***Wan et al., 2021***; ***Yamamoto et al., 2010***). In *Arabidopsis*, two genetically unlinked NLR proteins encoded on different chromosomes were shown to physically associate in the mixed immune background of hybrid plants, underpinning hybrid necrosis (***Tran et al., 2017***). Similarly, association between NLRs and alleles of non-NLR proteins derived from a different genetic background was also shown to induce NLR activation and autoimmune phenotypes (***Barragan et al., 2019***; ***Li et al., 2020***). However, the biochemical basis of adaptive specialization in genetically linked NLR receptor pairs remains largely unknown. In particular, we know little about how coevolution between paired NLRs has impacted their activities. We lack a validated framework to explain how plant immune receptors adapt and specialize, even though this process has important consequences for plant diversification and evolution (***Bomblies and Weigel, 2007***; ***Calvo-Baltanás et al., 2020***; ***Dobzhansky, 1937***; ***Li and Weigel, 2021***).

The rice NLRs *Pik-1* and *Pik-2* form a linked gene pair arranged in an inverted configuration on chromosome 11 with only ~2.5 kb separating their start codons (***Ashikawa et al., 2008***). The Pik pair is present in the genetic pool of rice cultivars as two major haplotypes (***Białas et al., 2021***; ***Kanzaki et al., 2012***). Pik pairs belonging to the K haplotype confer resistance to strains of the rice blast fungus, *Magnaporthe oryzae*, that harbour the effector AVR-Pik (***Ashikawa et al., 2008***). The sensor NLR Pik-1 binds AVR-Pik effectors through a heavy metal-associated (HMA) domain integrated into its architecture (***De la Concepcion et al., 2018***; ***Kanzaki et al., 2012***; ***Maqbool et al., 2015***). Upon effector recognition, Pik-1 cooperates with the helper NLR Pik-2 to activate immune signalling (***Zdrzałek et al., 2020***) that leads to pathogen resistance. The Pik NLR pair occurs as allelic series in both Japonica and Indica rice cultivars (***Chaipanya et al., 2017***; ***Costanzo and Jia, 2010***; ***Hua et al., 2012***; ***Xu et al., 2008***). The AVR-Pik effectors are also polymorphic and present signatures of selection (***Bentham et al., 2021***; ***Białas et al., 2018***; ***Yoshida et al., 2009***). Allelic Pik NLRs have differential recognition specificities for the AVR-Pik variants (***De la Concepcion et al., 2021***; ***Kanzaki et al., 2012***), which is underpinned by differential effector binding to the Pik-1 HMA domain (***De la Concepcion et al., 2018***; ***De la Concepcion et al., 2021***; ***Maqbool et al., 2015***). Two allelic variants of Pik-1, Pikp-1 and Pikm-1, acquired high-affinity binding to the *M. oryzae* AVR-Pik effector through

convergent evolution of their HMA domains (*Białas et al., 2021*). Additionally, Pikm-1 and Pikh-1 alleles have been shown to convergently evolve towards extended recognition specificity of AVR-Pik variants (*De la Concepcion et al., 2021*). This adaptive evolution towards recognition of rapidly evolving effectors has led to marked diversification of the integrated HMA domain (*Białas et al., 2021*). As a consequence, Pik-1 HMA domain is the most sequence-diverged domain in the Pik NLR pair (*Białas et al., 2021*; *Białas et al., 2018*; *Costanzo and Jia, 2010*).

While Pik-1 acts as a sensor, Pik-2 acts as a helper NLR that is required for the activation of immune responses (*Maqbool et al., 2015*; *Zdrzałek et al., 2020*). Evolutionary analyses have shown that the genetic linkage of this NLR pair is ancient and revealed marked signatures of adaptive evolution in the integrated HMA domain of Pik-1 (*Białas et al., 2021*). However, little is known about sensor/helper coevolution in Pik and how these multidomain proteins have adapted to changes in the rapidly evolving integrated HMA domain of Pik-1.

Here, we used two allelic variants of Pik, Pikp and Pikm, to explore NLR sensor/helper specificity (*Figure 1—figure supplement 1*). We challenged the hypothesis that throughout evolutionary time these two allelic Pik pairs have become diverged to the level of incompatibility. Indeed, mismatched pairs of Pik-1 and Pik-2 display constitutive cell death when combined in the heterologous system *Nicotiana benthamiana*, which is reminiscent of autoimmune phenotypes. We identified a single amino acid polymorphism in the helper NLR Pik-2 that underpins both allelic specialization and immune homeostasis. This finding allowed to reconstruct the evolutionary history of this coevolution. Altogether, these results demonstrate that NLR pairs can undergo co-adaptation and functional specialization, offering a molecular framework to understand how they evolve to respond to pathogen effectors while maintaining a tight regulation of immune responses.

## Results

### A coevolved Pik NLR pair is required for efficient cell death response to AVR-Pik effectors in *N. benthamiana*

Two of the most studied Pik alleles, Pikp (cv. K60) and Pikm (cv. Tsuyuake), fall into phylogenetically distinct groups (*Białas et al., 2021*; *De la Concepcion et al., 2021*; *Kanzaki et al., 2012*). Pikm originated in the Chinese Japonica cultivar Hokushi Tami (*Kiyosawa, 1978*) while Pikp originated in the Indica cultivar Pusur in Pakistan (*Kiyosawa, 1969*). Thus, we hypothesized that these alleles have been exposed to differential selection pressures during domestication of elite cultivars and have undergone distinct evolutionary trajectories.

To test for sensor/helper specificity in allelic Pik pairs, we co-expressed the sensor NLR Pikm-1 with either the helper NLR Pikp-2 or Pikm-2 in *N. benthamiana* and assessed the capacity to trigger a cell death in response to rice blast effector variants AVR-Pik D, E, or A (*Figure 1*). As previously reported, Pikm pair mediated a hierarchical cell death response in the order of AVR-PikD>AVR PikE>AVR PikA (*De la Concepcion et al., 2018*). However, the intensity of cell death was lower when Pikm-1 was co-expressed with Pikp-2 instead of Pikm-2 (*Figure 1*, *Figure 1—figure supplement 2*, *Figure 1—source data 1*). Protein accumulation of both Pikp-2 and Pikm-2 proteins in planta was similar (*Figure 1—figure supplement 3*).

These results indicate that Pikm-2 is required for the full Pikm-mediated cell death response to the AVR-Pik effectors in *N. benthamiana*. This suggests a possible functional specialization of the helper NLR Pik-2 towards an effective cell death response to these rice blast effectors.

### A single amino acid polymorphism in Pik-2 has an important role in cell death responses to the AVR-Pik effectors

To dissect the basis of the differential cell death phenotypes displayed by Pikp-2 and Pikm-2 in response to the AVR-Pik effectors, we used site-directed mutagenesis to exchange the residues at each of the only three Pik-2 polymorphic positions (*Figure 1—figure supplement 3*). We then co-expressed Pikm-1 and each of the Pik-2 mutants with either AVR-PikD, AVR-PikE, or AVR-PikA. For each assay, we scored the cell death responses and compared the differences with the Pikm control to qualitatively measure the contribution of each polymorphism to cell death (*Figure 2*, *Figure 2—figure supplement 1*, *Figure 2—figure supplement 2*, *Figure 2—figure supplement 3*, *Figure 2—figure*

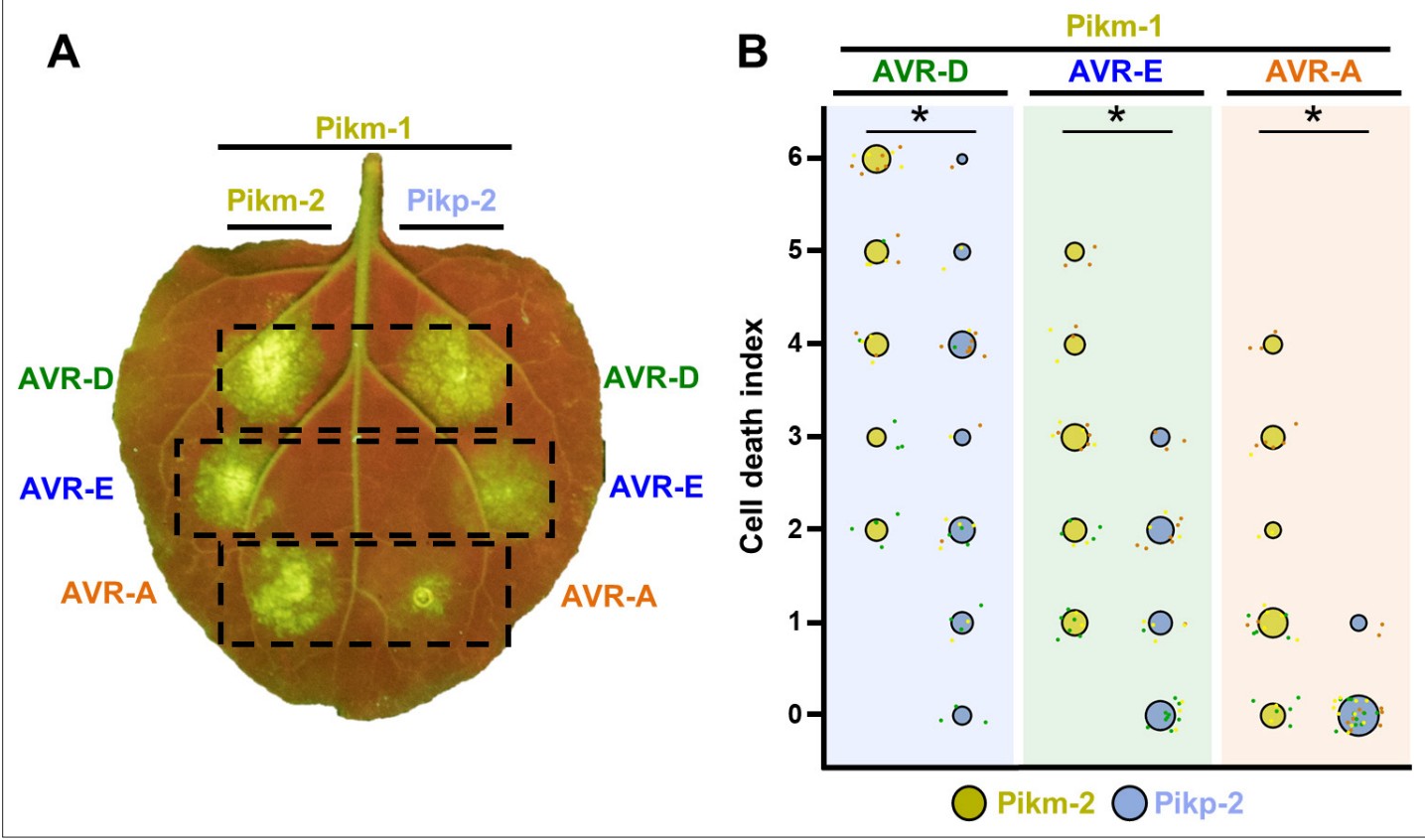

**Figure 1.** Pikm-1 elicits a stronger response to the AVR-Pik effectors when it is paired with Pikm-2 than with Pikp-2. (**A**) Representative *N. benthamiana* leaf depicting Pik-mediated cell death as autofluorescence under UV light. Pikm-1 was co-expressed with either Pikm-2 or Pikp-2 and the AVR-Pik effector alleles recognized by Pikm. Side-by-side infiltrations are highlighted with dashed boxes. (**B**) Scoring of cell death triggered by Pikp-2 or Pikm-2 with each AVR-PikD (AVR-D), AVR-PikE (AVR-E), and AVR-PikA (AVR-A) is represented as dot plots. The total number of repeats was 30. For each sample, all the data points are represented as dots with a distinct colour for each of the three biological replicates; these dots are jittered around the cell death score for visualization purposes. The size of the central dot at each cell death value is proportional to the number of replicates of the sample with that score. Significant differences between relevant conditions are marked with an asterisk, and the details of the statistical analysis are summarized in *Figure 1—figure supplement 2*.

The online version of this article includes the following figure supplement(s) for figure 1:

**Source data 1.** HR scores used for dot plots and statistics.

**Figure supplement 1.** Schematic representation of the hypothesis tested in this study.

**Figure supplement 2.** Estimation graphics for comparison of cell death mediated by Pikm-1 when co-expressed with Pikm-2 or Pikp-2.

**Figure supplement 3.** The Pik-2 alleles and mutants have similar levels of protein accumulation in planta.

*supplement 4*). In brief, this assay aimed to identify reciprocal mutations in Pikm-2 and Pikp-2 that may reduce or increase immune responses when compared with wild-type Pikm-2.

A single amino acid change at position 230 was responsible for the major differences in cell death responses (*Figure 2*). Despite the similar properties of their side chains, the Asp230Glu mutation in Pikp-2 showed an increase in the level of cell death response to AVR-Pik effectors (*Figure 2A*, *Figure 2—figure supplement 1A*, *Figure 2—source data 1*). By contrast, the Glu230Asp mutation in Pikm-2 reduced the cell death response to each AVR-Pik effector compared with wild-type Pikm-2, displaying only a slight response to AVR-PikD (*Figure 2B*, *Figure 2—figure supplement 1B*, *Figure 2—source data 2*). This points to a major involvement of the Pikm-2 Glu230 residue in the extended response to AVR-Pik effectors observed in Pikm (*De la Concepcion et al., 2018*).

Mutations at polymorphic positions 434 and 627 did not have the strong effect observed in the mutants at position 230. The Thr434Ser and Met627Val mutations in Pikp-2 did not yield higher levels of cell death response compared with Pikm-2 (*Figure 2—figure supplement 2A–C*, *Figure 2—figure supplement 3A–C*, *Figure 2—figure supplement 4A–C*, *Figure 2—figure supplement 3—source*

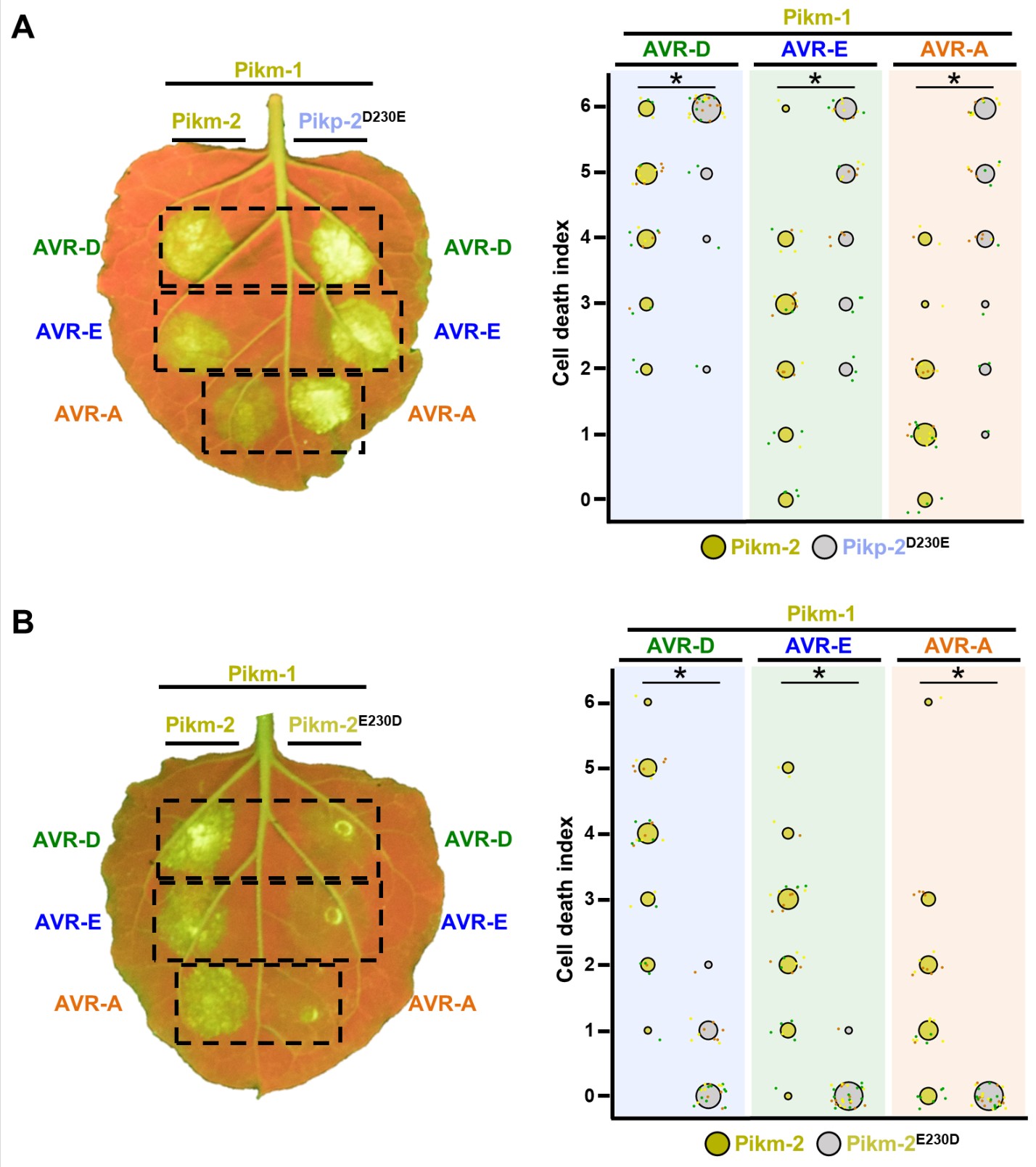

**Figure 2.** A single Pik-2 polymorphism modulates the cell death response to the AVR-Pik effectors. Representative leaves depicting cell death mediated by Pik-2 mutants as autofluorescence under UV light. Pikm-1 was co-expressed with either (**A**) Pikp-2 Asp230Glu or (**B**) Pikm-2 Glu230Asp and AVR-PikD (AVR-D), AVR-PikE (AVR-E), or AVR-PikA (AVR-A). Side-by-side infiltrations with Pikm NLR pair are highlighted with dashed boxes for comparison. Cell death scoring is represented as dot plots. The number of repeats was 30. For each sample, all the data points are represented as dots with a distinct

*Figure 2 continued on next page*

*Figure 2 continued*

colour for each of the three biological replicates; these dots are jittered about the cell death score for visualization purposes. The size of the central dot at each cell death value is proportional to the number of replicates of the sample with that score. Significant differences between relevant conditions are marked with an asterisk, and the details of the statistical analysis are summarized in *Figure 2—figure supplement 1*.

The online version of this article includes the following figure supplement(s) for figure 2:

**Source data 1.** HR scores used for dot plots and statistics (Pikm2 E230D).

**Source data 2.** HR scores used for dot plots and statistics (Pikm2 D230E).

**Figure supplement 1.** Estimation graphics for comparison of cell death mediated by Pikm-1 when co-expressed with the Pikm-2 or Pik-2 mutants in the polymorphic position 230.

**Figure supplement 2.** Representative images of cell death mediated by the Pik-2 mutants in response to the AVR-Pik effectors.

**Figure supplement 3.** Quantification of cell death mediated by the Pik-2 mutants in response to AVR-Pik effectors.

**Figure supplement 3—source data 1.** HR scores used for dot plots and statistics (Pikm2 S434T).

**Figure supplement 3—source data 2.** HR scores used for dot plots and statistics (Pikm2 V627M).

**Figure supplement 3—source data 3.** HR scores used for dot plots and statistics (Pikp2 M627V).

**Figure supplement 3—source data 4.** HR scores used for dot plots and statistics (Pikp2 T434S).

**Figure supplement 4.** Estimation graphics for comparison of cell death mediated by Pikm-1 when co-expressed with Pikm-2 or Pik-2 mutants in polymorphic position 434 and 627.

*data 3*, *Figure 2—figure supplement 3—source data 4*). Likewise, neither Pikm-2 Ser434Thr nor Pikm-2 Val627Met showed a lower level of cell death response compared with wild-type Pikm-2 (*Figure 2—figure supplement 2B–D*, *Figure 2—figure supplement 3B–D*, *Figure 2—figure supplement 4B–D*, *Figure 2—figure supplement 3—source data 1*, *Figure 2—figure supplement 3—source data 2*). Interestingly, Val627Met in Pikm-2 consistently increased cell death responses, particularly to AVR-PikE and AVR-PikA (*Figure 2—figure supplement 2D*, *Figure 2—figure supplement 3D*, *Figure 2—figure supplement 4D*) implying a negative contribution of Pikm-2 polymorphism Val627 towards cell death responses. All mutants had a similar level of protein accumulation in *N. benthamiana* compared to wild-type Pikp-2 and Pikm-2 (*Figure 1—figure supplement 3*).

Altogether, these results demonstrate that polymorphisms in Pik-2 play an important role in facilitating response to different AVR-Pik alleles. Particularly, a single polymorphic residue, Glu230, was revealed as a major determinant of the increased cell death responses to the AVR-Pik effectors displayed by the Pikm NLR pair.

## Mismatched Pik pair Pikp-1/Pikm-2 triggers constitutive cell death responses in *N. benthamiana*

When independently evolved NLR receptors meet in the mixed immune background of a hybrid plant, it can lead to misregulation in the form of suppression (*Hurni et al., 2014*; *Stirnweis et al., 2014*) or constitutive activation of immune responses (*Chae et al., 2014*; *Li et al., 2020*; *Tran et al., 2017*).

The Pikp and Pikm allelic pairs trigger a strong cell death response in *N. benthamiana* when co-expressed with rice blast effector AVR-PikD, but not in the absence of effector (*De la Concepcion et al., 2018*; *Maqbool et al., 2015*). However, we noticed that when Pikp-1 was co-expressed together with Pikm-2, it led to cell death response in the absence of AVR-PikD (*Figure 3*, *Figure 3—source data 1*). We did not observe NLR autoactivation in the reciprocal mismatched pair Pikm-1/Pikp-2 (*Figure 3*, *Figure 3—source data 1*).

These results reveal signatures of coevolution in the Pikp and Pikm allelic pairs. We hypothesize that these allelic pairs have coevolved with their respective partners and have drifted enough to trigger a misregulated form of immune response when they are mismatched, leading to constitutive cell death in *N. benthamiana*.

## Pik autoactivity is linked to immune signalling

We sought to gain knowledge on the constitutive cell death mediated by Pikm-2 and understand the link with NLR activation. To this end, we mutated Pikm-2 in the conserved P-loop and MHD motifs and tested their ability to trigger constitutive cell death responses in the absence of the AVR-PikD effector.

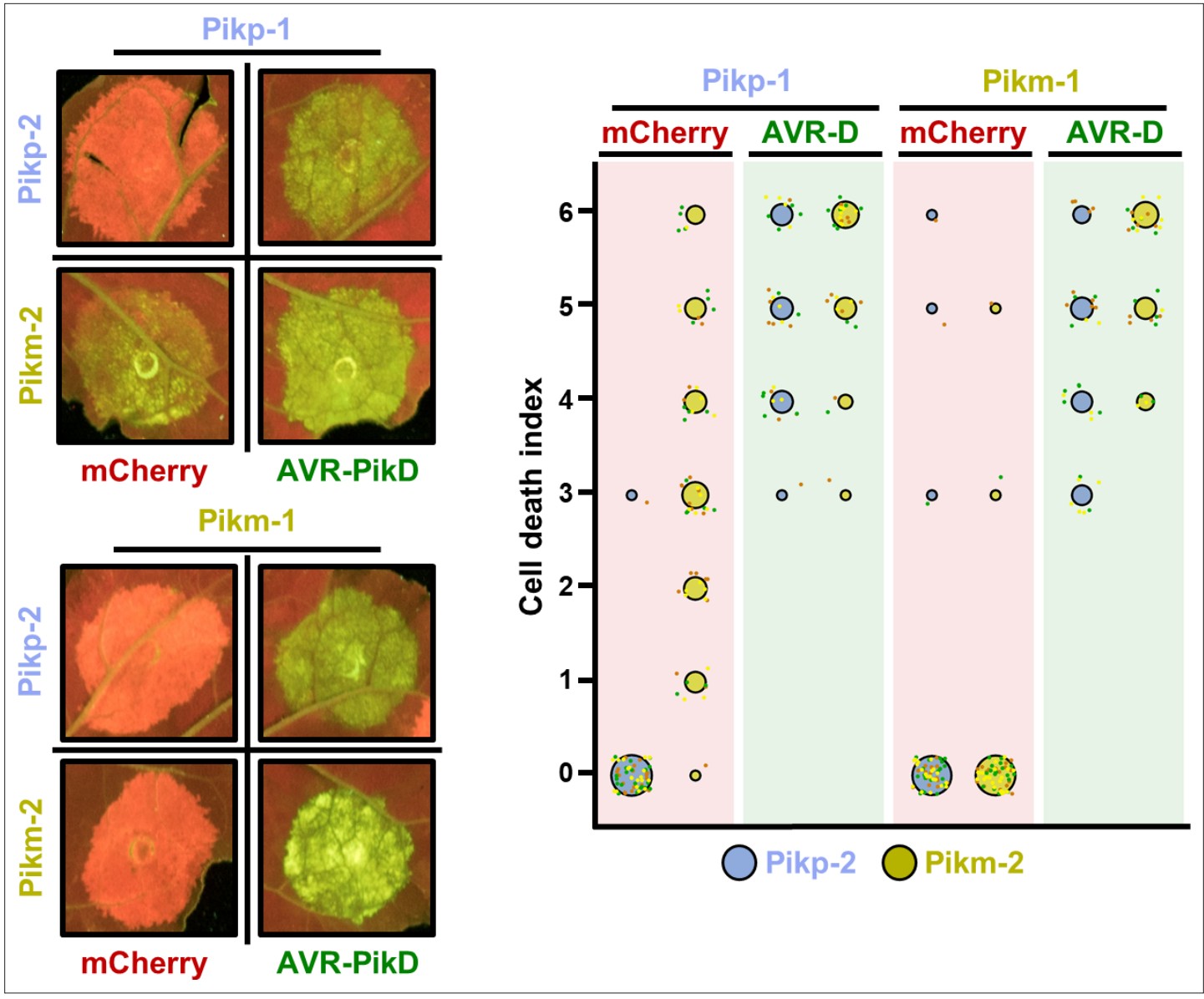

**Figure 3.** Pikm-2 triggers constitutive cell death in the presence of Pikp-1. Representative leaf spot images and scoring of Pik-mediated cell death as autofluorescence under UV light in the presence or absence of AVR-PikD. Cell death assay scoring represented as dot plots comparing cell death triggered by Pikp-2 and Pikm-2 when co-expressed with Pikp-1 or Pikm-1. The number of repeats was 60 and 30 for the spots co-infiltrated with mCherry and AVR-PikD, respectively. For each sample, all the data points are represented as dots with a distinct colour for each of the three biological replicates; these dots are jittered about the cell death score for visualization purposes. The size of the central dot at each cell death value is proportional to the number of replicates of the sample with that score.

The online version of this article includes the following figure supplement(s) for figure 3:

**Source data 1.** HR scores used for dot plots.

The P-loop motif is conserved in NLR proteins and mediates nucleotide binding linked with oligomerization and NLR activation (*Ma et al., 2020*; *Wang et al., 2019b*). Loss-of-function mutations at this position render NLRs inactive and have been extensively documented (*Tameling et al., 2002*; *Tameling et al., 2006*; *Williams et al., 2011*). A Lys217Arg mutation in the P-loop motif of Pikp-2 abrogates Pik-mediated cell death responses to AVR-PikD in *N. benthamiana* (*Zdrzałek et al., 2020*). Introducing this mutation in Pikm-2 abolished Pikm-mediated cell death response to the rice blast effector AVR-PikD (*Figure 4*, *Figure 4—source data 1*) and also abrogated the constitutive cell death response triggered by the Pikp-1/Pikm-2 NLR mismatch (*Figure 4*, *Figure 4—source data 1*).

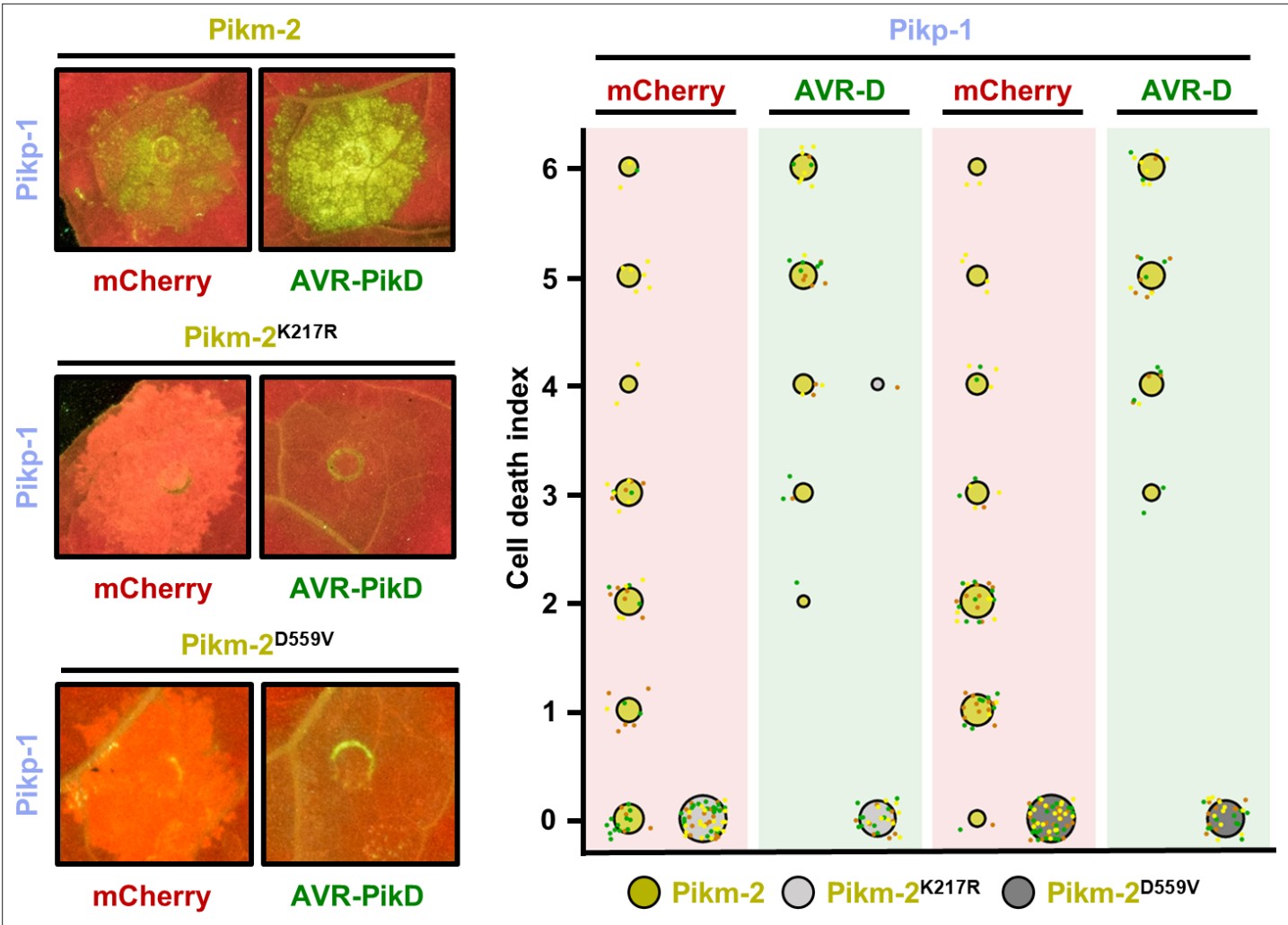

**Figure 4.** Constitutive cell death in mismatched Pik pairs is dependent on P-loop and MHD motifs. Representative leaf spot images and scoring of Pikm-2-mediated cell death as autofluorescence under UV light. Cell death scoring is represented as dot plots comparing cell death triggered by Pikm-2 mutant in P-loop (Lys217Arg) and MHD (Asp559Val) motifs and wild-type Pikm-2. Mutants and wild-type proteins were co-expressed with Pikp-1 and mCherry (red panel) or AVR-PikD (green panel). The number of repeats was 60 and 30 for the spots co-infiltrated with mCherry and AVR-PikD, respectively. For each sample, all the data points are represented as dots with a distinct colour for each of the three biological replicates; these dots are jittered about the cell death score for visualization purposes. The size of the central dot at each cell death value is proportional to the number of replicates of the sample with that score.

The online version of this article includes the following figure supplement(s) for figure 4:

**Source data 1.** HR scores used for dot plots (P-loop).

**Source data 2.** HR scores used for dot plots (MHD).

NLR activities are also altered by mutations in the MHD motif. An Asp to Val mutation in this motif is predicted to change ATP/ADP binding preference and, in many cases, renders NLRs constitutively active (*Bernoux et al., 2016*; *Tameling et al., 2006*; *Williams et al., 2011*). Contrary to other NLRs, introducing Asp559Val in the MHD motif of Pikp-2 abolished cell death responses to AVR-PikD (*Zdrzałek et al., 2020*). Consequently, we introduced the equivalent mutation in Pikm-2 and verified that it also abrogated cell death in autoimmune combinations (*Figure 4*, *Figure 4—source data 2*), confirming that Pikm-2 requires an intact MHD motif to trigger cell death and strengthening the link between constitutive cell death and immune activation.

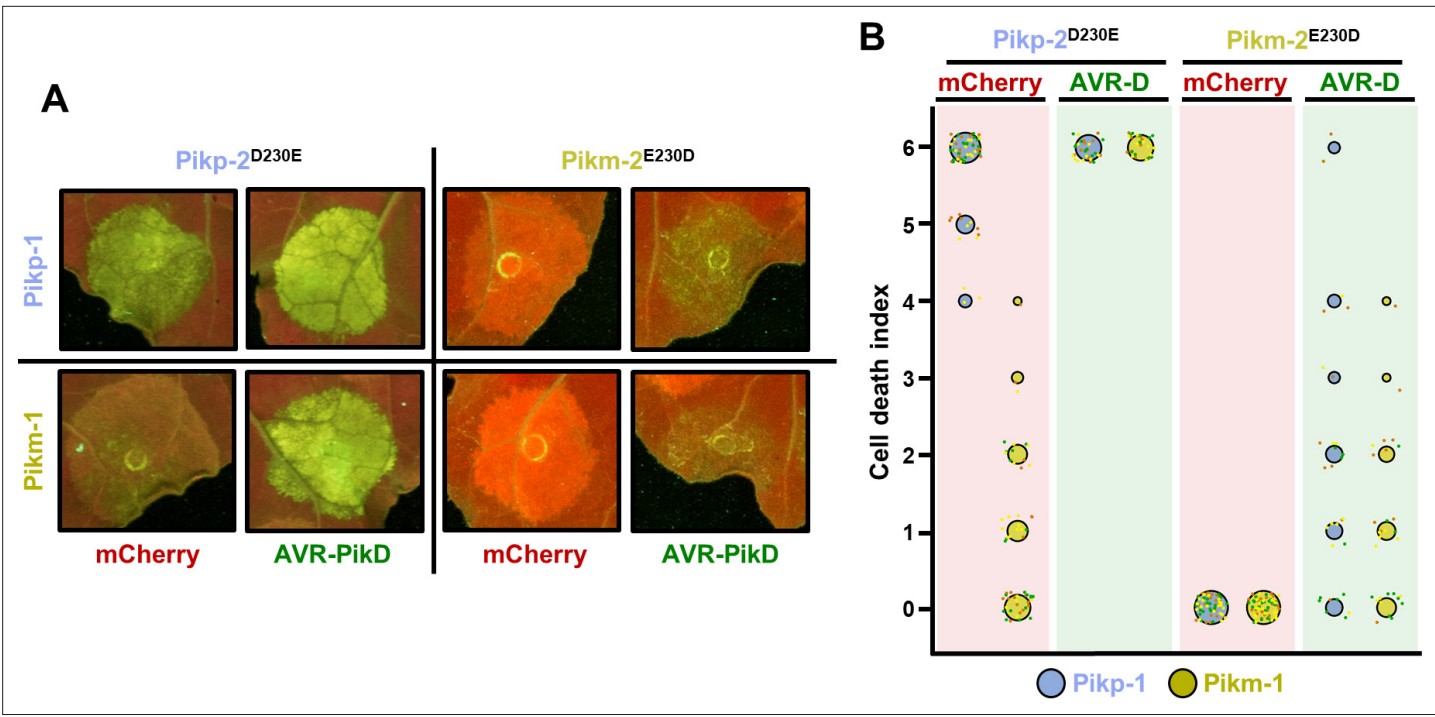

**Figure 5.** Polymorphism at position 230 in the NB-ARC domain is a Pik-2 determinant for constitutive cell death. (**A**) Representative leaf spot images and scoring of cell death mediated by Pik-2 as autofluorescence under UV light. (**B**) Cell death scoring is represented as dot plots comparing cell death triggered by Pik-2 mutants at polymorphic positions 230. Pik-2 mutants were co-expressed with Pikp-1 (blue dots) or Pikm-1 (yellow dots) together with mCherry (red panel) or AVR-PikD (green panel). The number of repeats was 60 and 30 for the spots co-infiltrated with mCherry and AVR-PikD, respectively. For each sample, all the data points are represented as dots with a distinct colour for each of the three biological replicates; these dots are jittered about the cell death score for visualization purposes. The size of the central dot at each cell death value is proportional to the number of replicates of the sample with that score.

The online version of this article includes the following figure supplement(s) for figure 5:

**Source data 1.** HR scores used for dot plots.

**Figure supplement 1.** The Pik-2 polymorphisms at positions 434 and 627 do not alter constitutive cell death.

**Figure supplement 1—source data 1.** HR scores used for dot plots.

**Figure supplement 1—source data 2.** HR scores used for dot plots.

**Figure supplement 2.** Pikp-2 Asp230Glu autoactivation is dependent on P-loop and MHD motifs.

**Figure supplement 2—source data 1.** HR scores used for dot plots.

**Figure supplement 3.** The Pik-1 P-loop motif is important but not essential for Pik-mediated cell death.

**Figure supplement 3—source data 1.** HR scores used for dot plots.

**Figure supplement 4.** Mutations in P-loop and MHD motifs do not affect protein accumulation.

## NLR specialization and autoimmunity are linked to the same amino acid polymorphism

Interestingly, only mismatches involving Pikm-2 triggered cell death in the absence of the effector (*Figure 3*), suggesting that this NLR harbours the determinants of this autoactive phenotype. To understand the basis of Pikm-2-mediated autoimmunity, we used the point mutants in Pik-2 polymorphic positions presented above (*Figure 1—figure supplement 3*) to explore the determinant of constitutive cell death. To this end, we co-expressed each mutant with either Pikp-1 or Pikm-1 in the presence or absence of AVR-PikD effector (*Figure 5*, *Figure 5—figure supplement 1*). In this assay, we added AVR-PikD effectors as a positive control for cell death.

The Asp230Glu mutation in Pikp-2 conferred a strong cell death response in the absence of the effector when co-expressed with Pikp-1, while only residual constitutive activation could also be observed with Pikm-1 (*Figure 5*, *Figure 5—source data 1*). By contrast, the reciprocal mutation at the equivalent position in Pikm-2 abrogated constitutive cell death in the presence of Pikp-1 and

reduced the cell death response mediated by AVR-PikD recognition (*Figure 5*, *Figure 5—source data 1*). Single mutations in any of the other polymorphic positions had no effect on constitutive cell death activation (*Figure 5—figure supplement 1*, *Figure 5—figure supplement 1—source data 1*, *Figure 5—figure supplement 1—source data 2*).

Additionally, we confirmed that constitutive cell death triggered by Pik-2 Asp230Glu is also dependent on the P-loop and MHD motifs, confirming that this mutation leads to immune activation (*Figure 5—figure supplement 2*, *Figure 5—figure supplement 2—source data 1*). Interestingly, cell death responses were reduced but not completely abolished when Pikm-2 or Pikp-2 Asp230Glu were co-expressed with a P-loop mutant of Pikp-1 (*Figure 5—figure supplement 3*, *Figure 5—figure supplement 3—source data 1*). Protein accumulation of the Pikp-1 P-loop mutant and the Pik-2 P-loop and MHD mutants was equivalent to their wild-type counterparts (*Figure 5—figure supplement 4*). This unequal contribution of the P-loop motifs of sensor and helper NLRs adds an extra layer of information to the cooperation model of NLR activation previously proposed for Pik (*Białas et al., 2018*).

Overall, we narrowed down a determinant of autoimmunity in the mismatched Pik pairs to a single amino acid polymorphism. Furthermore, we confirmed that this polymorphism mediates cell death phenotypes by a mechanism dependent on the P-loop and MHD motifs. Interestingly, the same polymorphism is related to the stronger cell death responses to AVR-Pik effectors mediated by Pikm compared to Pikp (*Figure 2*). Altogether, this establishes a link between immune specialization and gain of constitutive cell death responses in NLR pairs, two hallmarks of coevolution.

## The Glu230 amino acid polymorphism has evolved in modern rice

Having identified a determinant of Pik NLR pair specialization and compatibility as a single amino acid polymorphism, we aimed to gain an evolutionary perspective of the specialization process of Pik-2. For this, we combined the Pik-2 coding sequences from rice cultivars described above with the Pik-2 orthologs from wild Asian and African relative species (*Białas et al., 2021*; *Stein et al., 2018*) (see Materials and methods for accession numbers) and calculated the maximum likelihood phylogenetic tree rooted in the African outgroup species *Leersia perrieri* (*Figure 6A*).

Pik-2 sequences from wild rice species are phylogenetically distinct from those belonging to modern rice, with the exception of Nipponbare (*Figure 6A*). These modern varieties make two distinct groups harbouring Pikp cultivar K60 or Pikm cultivar Tsuyuake (*Figure 6A*; *Białas et al., 2021*; *De la Concepcion et al., 2021*).

To learn more of the evolutionary trajectory of Pik-2, we inferred the ancestral state of the nucleotide sequences coding for the polymorphic position 230. This analysis revealed that a Gly residue encoded by GGT is an ancestral state at this position and is still present in most Pik-2 sequences from wild *Oryza* species (*Figure 6A*).

A transition from GGT (coding for Gly) to GAT (coding for Asp) in position 230 occurred before the split of *Oryza sativa* and *Oryza punctata* and has been maintained in Pik-2 NLRs of modern rice varieties clustering in the Pikp cultivar K60 (*Figure 6A*). This change opened the possibility of a non-synonymous Asp to Glu mutation by a GAT to GAA transversion, which occurred in the rise of the clade containing the Pikm cultivar Tsuyuake. This Asp230Glu polymorphism represents a specialization determinant in the Pikm NLR pair and ultimately rendered Pikm-2 incompatible with Pikp-1.

To experimentally validate the reconstructed evolutionary history of Pik-2 polymorphic position 230, we reverted this position in Pikm-2 to the ancestral state by introducing a Glu230Gly mutation and tested its ability to trigger cell death in *N. benthamiana*. The Glu230Gly mutation abolished the constitutive cell death triggered by Pikm-2 when co-expressed with Pikp-1 in the absence of the effector (*Figure 6B*). This mutation did not abrogate the cell death response to the AVR-PikD effector, although it slightly reduced it compared with the wild type (*Figure 6B*, *Figure 6—figure supplement 1*, *Figure 6—figure supplement 1—source data 1*). Protein accumulation of Pikm-2 Glu230Gly was equivalent to wild-type Pikp-2 and Pikm-2 (*Figure 6—figure supplement 2*).

Overall, having reconstructed the evolutionary history of Pik NLR specialization we propose a model where a multi-step mutation led to the emergence of Glu230 polymorphism, which is linked to an efficient cell death response to AVR-Pik effectors in the Pikm pair. We further demonstrated that the rise of this polymorphism is associated with NLR incompatibility with mismatched sensor NLRs

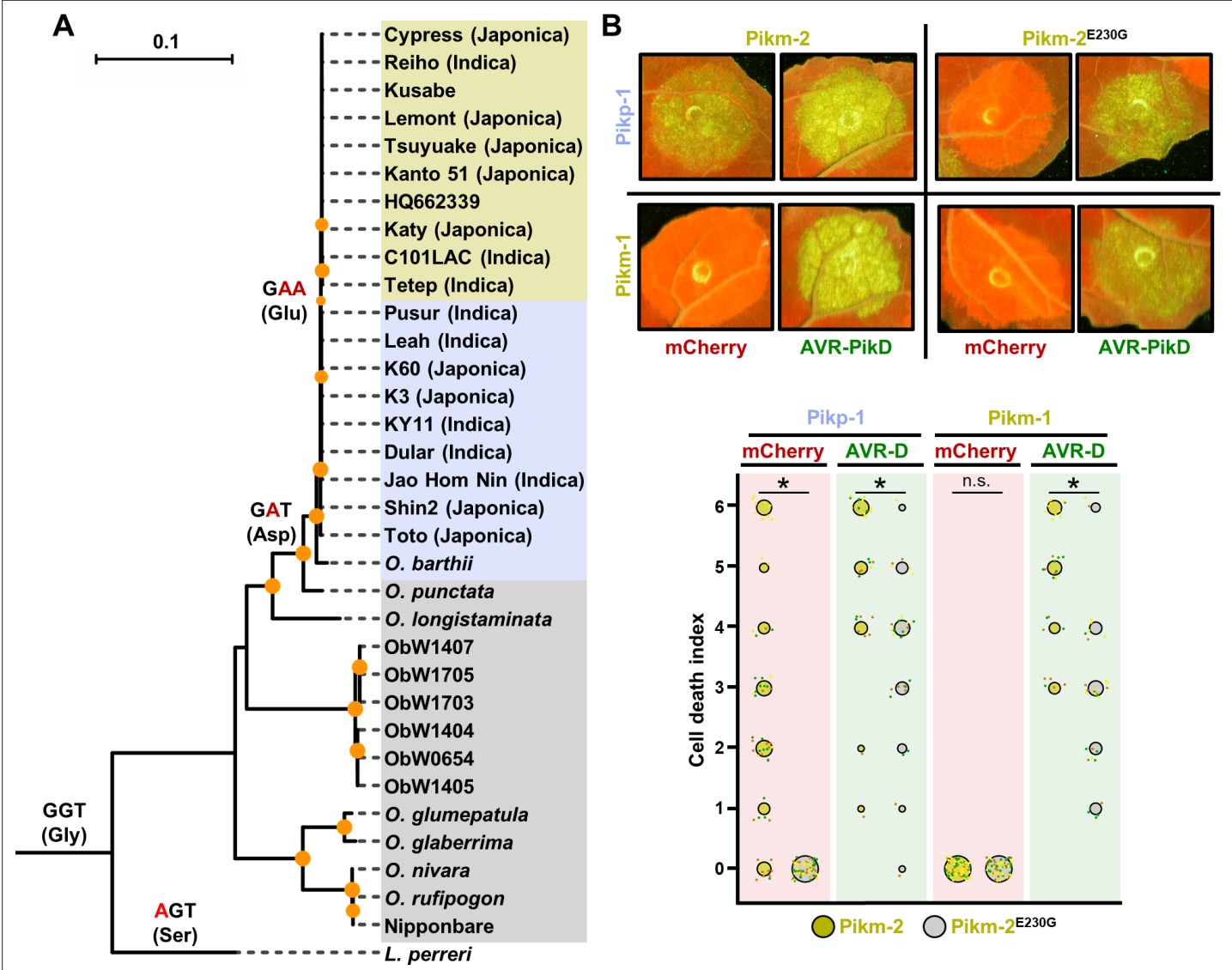

**Figure 6.** The Glu230 amino acid polymorphism has evolved in modern rice. (**A**) Reconstruction of the evolutionary history of Pik-2 polymorphism at position 230.Maximum likelihood (ML) phylogenetic tree of Pik-2 coding sequences from cultivated rice and wild rice species. The tree was calculated from a 3066-nt-long alignment using RAxML v8.2.11 (**Stamatakis, 2014**), 1000 bootstrap method (**Felsenstein, 1985**) and GTRGAMMA substitution model (**Tavaré, 1986**). Best-scoring ML tree was manually rooted using the Pik-2 sequence from *Leersia perreri* as an outgroup. The bootstrap values above 80 are indicated with orange circles at the base of respective clades; the support values for the relevant nodes are depicted by the size of the circle. The scale bar indicates the evolutionary distance based on the nucleotide substitution rate. The tree is represented using Interactive Tree Of Life (iTOL) v4 (**Letunic and Bork, 2019**). The tree shows a set of inferred nucleotides (states) at the Pik-2 polymorphic position 230 based on their predicted likelihood at sites 709–711 of the sequence alignment. Non-synonymous changes at the codon are depicted in red next to their corresponding node. For visualization, rice species and cultivars names are shaded in gold, light blue, or grey according to their residue in Pik-2 polymorphic position 230 (Glu, Asp, or Gly, respectively). For modern rice cultivars, it is indicated in brackets whether they are Japonica or Indica variety (when known). Ob: *Oryza brachyantha*. (**B**) Reversion to ancestral state of Pikm-2 Glu230 abolishes autoimmunity. Representative leaf spot images depicting Pik-mediated cell death as autofluorescence under UV light in the presence or absence of AVR-Pik effector. Scoring of the cell death triggered by Pikm-2 or Pikm-2 Glu230Gly mutant when co-expressed with Pikp-1 or Pikm-1 is represented as dot plots. The number of repeats was 60 and 30 for the spots co-infiltrated with mCherry and AVR-PikD, respectively. For each sample, all the data points are represented as dots with a distinct colour for each of the three biological replicates; these dots are jittered about the cell death score for visualization purposes. The size of the central dot at each cell death value is proportional to the number of replicates of the sample with that score. Significant differences between relevant conditions are marked with an asterisk, and the details of the statistical analysis are summarized in *Figure 6—figure supplement 1*.

The online version of this article includes the following figure supplement(s) for figure 6:

**Figure supplement 1.** Estimation graphics for comparison of cell death mediated by Pikm-2 or Pikm-2 Glu230Gly.

*Figure 6 continued on next page*

*Figure 6 continued*

**Figure supplement 1—source data 1.** HR scores used for dot plots and statistics.

**Figure supplement 2.** Glu230Gly mutation does not affect Pik-2 protein accumulation.

from the Pikp-like clade, triggering constitutive immune activation and cell death in the absence of pathogen effectors.

## Sensor/helper hetero-pairing alters protein accumulation in Pik NLRs

We aimed to obtain mechanistic understanding of Pik NLR pair coevolution and autoactivation. For this, we investigated whether accumulation of sensor Pik-1 or helper Pik-2 proteins is altered in the presence of the coevolved or mismatched pair.

After co-expression of both Pikp-1 and Pikm-1 alleles in *N. benthamiana* in combination with the helper Pikp-2 or Pikm-2 alleles followed by western blot, we observed that protein accumulation of Pik-1 and Pik-2 alleles was consistently increased when they were expressed together compared to co-expression with empty vector (**Figure 7A**). This is consistent with a model where Pik-1 and Pik-2 associate in sensor/helper NLR heterocomplexes, stabilizing both proteins (**Zdrzałek et al., 2020**).

Interestingly, accumulation of the helper Pik-2 in the autoimmune pair Pikp-1/Pikm-2 was consistently higher (**Figure 7A**). This could be due to a different sensor/helper stoichiometry in the constitutively active Pik complex, as observed in some activated NLR complexes (**Hu et al., 2015**; **Sharif et al., 2019**; **Tenthorey et al., 2017**; **Zhang et al., 2015**). This is also consistent with the finding that

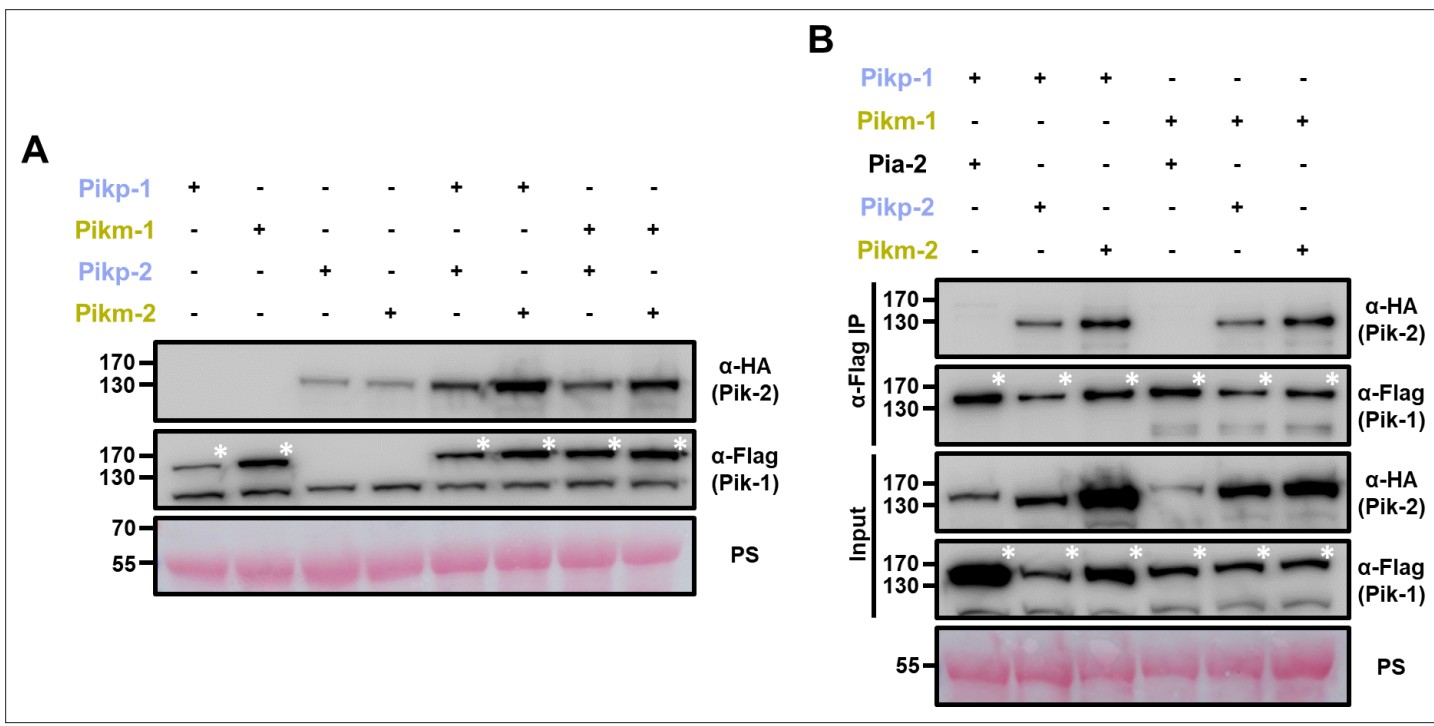

**Figure 7.** Coevolved and mismatched Pik pairs form heterocomplexes. (**A**) Increased protein accumulation of paired Pik proteins when co-expressed together in planta. Western blots showing protein accumulation of Pik-1 and Pik-2 alleles in different combinations. C-terminally 6×His/3×FLAG tagged Pik-1 alleles were transiently co-expressed with empty vector (EV) or C-terminally 6×HA tagged Pik-2 alleles in *N. benthamiana*. Total protein extracts were probed with α-FLAG and α-HA antisera for Pik-1 and Pik-2, respectively. Asterisks mark the band corresponding to Pik-1. (**B**) Mismatched Pik NLR pairs associate in planta. Co-immunoprecipitation of full-length Pikp-1 and Pikm-1 alleles in combination with either Pikp-2 or Pikm-2 helper NLRs. C-terminally 6×HA tagged Pia-2, Pikp-2, or Pikm-2 NLRs were transiently co-expressed with Pikp-1: 6×His/3×FLAG or Pikm-1:6×His/3×FLAG in *N. benthamiana*. Immunoprecipitates obtained with anti-FLAG antiserum, and total protein extracts, were probed with appropriate antisera. Asterisks mark the band corresponding to Pik-1. Total protein loading is shown by Ponceau staining (PS).

The online version of this article includes the following figure supplement(s) for figure 7:

**Figure supplement 1.** Pik-2 mutants associate with Pik-1 in planta.

**Figure supplement 2.** Reversion to ancestral state in polymorphism 230 does not abrogate association with Pik-1 alleles.

CC domain of Pik-2 NLR has the consensus MADA motif first identified in ZAR1 (*Adachi et al., 2019a*), indicating the possibility that Pik activation may involve oligomerization of multiple Pik-2 receptors as in the ZAR1 resistosome (*Wang et al., 2019a*).

## Coevolved and mismatched Pik pairs form heterocomplexes

Prompted by the differences in protein accumulation observed between different combinations of Pik-1 and Pik-2, we investigated whether cell death phenotypes in mismatched Pik pairs are underpinned by differences in NLR hetero-association.

We co-expressed C-terminally tagged Pikp-1 or Pikm-1 with either C-terminally tagged Pikp-2 or Pikm-2 in *N. benthamiana*. Following total protein extraction, we performed co-immunoprecipitation (Co-IP) to test for differences in NLR association (*Figure 7B*). Pikp-1 and Pikm-1 were also co-infiltrated with the rice NLR Pia-2 (the sensor NLR, also known as RGA5, of the immune receptor pair Pia) as a negative control.

Both Pikp-2 and Pikm-2 could be detected after immunoprecipitation of either Pikp-1 or Pikm-1 sensor NLRs (*Figure 7B*). Additionally, none of the Pik-2 mutations generated above seem to have a measurable effect on the sensor/helper association (*Figure 7—figure supplement 1*, *Figure 7—figure supplement 2*).

These results indicate that cell death phenotypes observed in mismatched pairs are not underpinned by major alterations in association. Instead, Pik sensor and helper NLRs may form pre-activation complexes in the resting state and subtle changes, perhaps both in association and stoichiometry between Pik NLRs, govern cell death responses and autoimmune phenotypes described above.

## Sensor/helper association of Pik NLR pairs is independent of NLR activation

As Pik NLR pairs associate in pre-activation complexes (*Figure 7B*; *Zdrzałek et al., 2020*), we investigated whether this process requires functional NLRs. We co-expressed the Pikm-2 P-loop and MHD mutants with either Pikp-1 or Pikm-1 in *N. benthamiana*. Following protein extraction and immunoprecipitation of Pik-1, we found that these mutations do not affect the ability to associate with the sensor NLR Pik-1 compared to wild-type Pikm-2 (*Figure 8*), although they completely abolish Pik-mediated cell death. Similarly, the reduced cell death activity in the Pik-1 P-loop mutant did not correlate with alterations in the association to the helper NLR Pik-2 (*Figure 8—figure supplement 1*).

These results imply that pre-activated Pik NLR pair association does not require functional NLRs and is independent of nucleotide binding. In the native state, such pre-activation complexes may require ADP/ATP exchange to induce or stabilize changes in receptor conformation and/or stoichiometry to trigger immune signalling.

## Sensor and helper Pik NLRs preferentially associate with their coevolved pair

To gain a deeper knowledge of Pik pair association, we investigated whether allelic Pik NLRs display any preference in association to their coevolved NLR pair. As both autoactive and non-autoactive pairs associate, we designed an NLR competition assay with a cell death readout to test for preferential association between allelic NLRs (*Figure 9—figure supplement 1*). For this, we took advantage of the constitutive cell death phenotype triggered by the association of Pikp-1 and Pikm-2 (*Figure 9—figure supplement 1A*). In a scenario where a non-autoactive Pik-2 NLR displays higher helper/sensor association to Pikp-1, Pikm-2 would be outcompeted from complex formation, reducing the levels of constitutive cell death (*Figure 9—figure supplement 1B*).

To test this, we transiently co-expressed both Pikp-1 and Pikm-2 NLRs in *N. benthamiana* using a fixed concentration ($OD_{600}$ 0.4) of *Agrobacterium tumefaciens* to deliver each construct. We also co-delivered increasing concentrations of Pikp-2 (spanning an $OD_{600}$ of 0–0.6) and scored the cell death phenotype (*Figure 9*).

Interestingly, Pikp-2 acted as a suppressor of autoimmune phenotypes triggered by Pikp-1/Pikm-2 as increasing concentrations of Pikp-2 lowered the constitutive cell death phenotype (*Figure 9A and B*, *Figure 9—source data 1*). This reduction in cell death was evident even in the lowest concentration of Pikp-2 (*Figure 9A and B*, *Figure 9—source data 1*), suggesting that Pikp-1 displays preference to signal through coevolved Pikp-2 rather than Pikm-2.

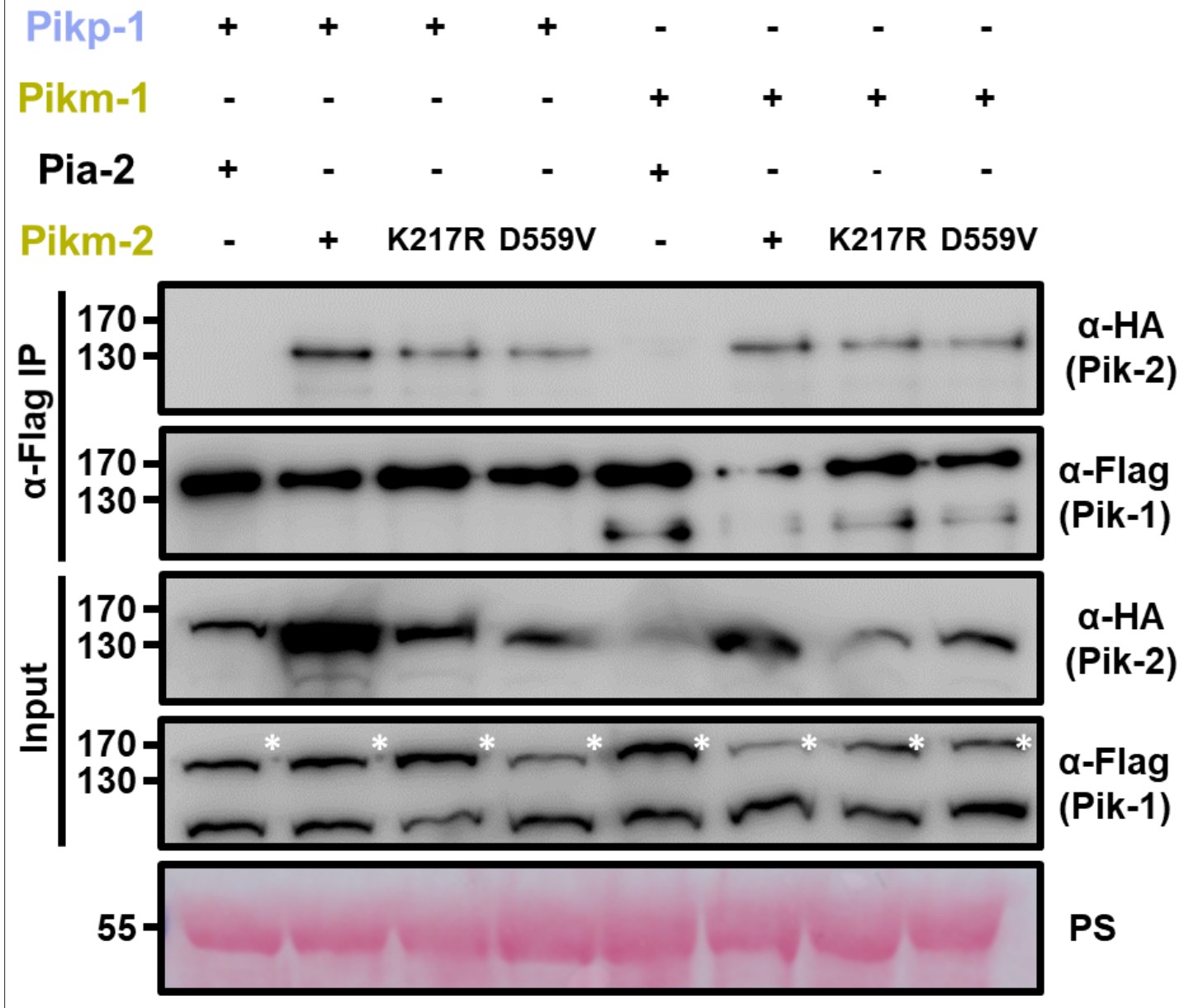

**Figure 8.** Mutations in Pik-2 P-loop and MHD motifs do not affect in planta association of Pik-1. Co-immunoprecipitation of Pikm-2 P-loop and MHD mutants with full-length Pikp-1 and Pikm-1 alleles. C-terminally 6×HA tagged Pikm-2 mutants in P-loop (Lys217Arg) and MHD (Asp559Val) motifs were transiently co-expressed with either Pikp-1: 6×His/3×FLAG or Pikm-1:6×His/3×FLAG in *N. benthamiana*. Immunoprecipitates obtained with anti-FLAG antiserum, and total protein extracts, were probed with appropriate antisera. Co-expression with C-terminally tagged 6×HA Pia-2 NLR and wild-type Pikm-2 is included as negative and positive control, respectively. Asterisks mark the band corresponding to Pik-1. Total protein loading is shown by Ponceau staining (PS).

The online version of this article includes the following figure supplement(s) for figure 8:

**Figure supplement 1.** P-loop mutations do not affect Pik-1 association to Pik-2.

We also replicated this experiment co-infiltrating a fixed concentration of Pikp-1 and Pikp-2, with increasing concentration of Pikm-2. In agreement with a signalling preference between Pikp-1 and Pikp-2, Pikm-2 could not overcome the suppression by the presence of Pikp-2, even at the highest concentration (***Figure 9—figure supplement 2***).

To investigate whether the decrease in cell death is correlating with reduced association of the maladapted pair Pikp-1/Pikm-2 in the presence of Pikp-2, we immunoprecipitated Pikp-1 and tested for the presence of Pikp-2 or Pikm-2 (***Figure 9C***).

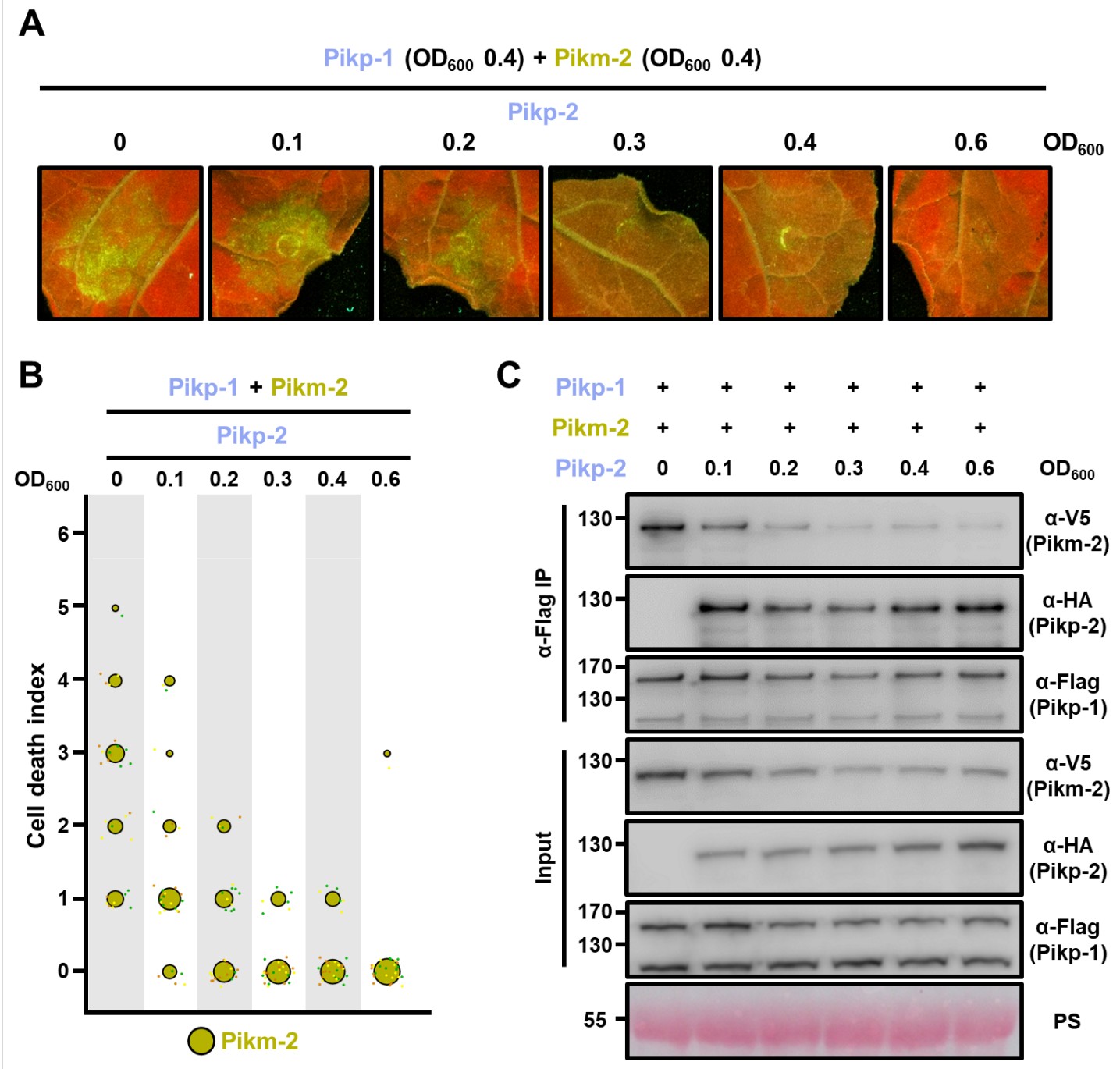

**Figure 9.** Pikp-2 supresses constitutive cell death mediated by Pikm-2. (**A**) Representative leaf spot images depicting Pikm-2-mediated cell death in the presence of Pikp-1 and increasing concentration of Pikp-2 as autofluorescence under UV light. For each experiment, Pikp-1 and Pikm-2 were co-infiltrated at $OD_{600}$ 0.4 each. Increasing concentrations of Pikp-2 were added to each experiment (from left to right: $OD_{600}$ 0, 0.1, 0.2, 0.3, 0.4, and 0.6). (**B**) Scoring of the cell death assay is represented as dot plots. A total of three biological replicates with 10 internal repeats each were performed for each experiment. For each sample, all the data points are represented as dots with a distinct colour for each of the biological replicates; these dots are jittered about the cell death score for visualization purposes. The size of the central dot at each cell death value is proportional to the number of replicates of the sample with that score. (**C**) Pikp-2 outcompetes Pikm-2 association to Pikp-1. Co-immunoprecipitation of Pikm-2 and Pikp-1 in the presence of increasing concentrations on Pikp-2. C-terminally V5 tagged Pikm-2 and C-terminally 6×His/3×FLAG tagged Pikp-1 were transiently co-expressed in *N. benthamiana* alongside with increasing concentrations of C-terminally 6×HA tagged Pikp-2 (from left to right: 0, 0.1, 0.2, 0.3, 0.4, and 0.6 $OD_{600}$). Immunoprecipitates obtained with anti-FLAG antiserum, and total protein extracts, were probed with appropriate antisera. Asterisks mark the band corresponding to Pikp-1. Total protein loading is shown by Ponceau staining (PS).

*Figure 9 continued on next page*

*Figure 9 continued*

The online version of this article includes the following source data and figure supplement(s) for figure 9:

**Source data 1.** HR scores used for dot plots.

**Figure supplement 1.** Schematic representations of Pik NLR competition assays.

**Figure supplement 2.** Pikp-2 suppresses constitutive cell death mediated by Pikm-2.

**Figure supplement 2—source data 1.** HR scores used for dot plots.

Differences in protein accumulation observed in the different sensor/helper combinations of Pik pairs make it particularly challenging to obtain even inputs for this experiment. As reported above (*Figure 7A*), the Pik-2 proteins are more stable in association with Pik-1; therefore, if a Pik-2 protein is outcompeted from a hypothetical complex, it will present reduced accumulation in the input. The contrary effect occurs in the Pik-2 proteins forming autoactive complexes; as they showed increased accumulation in autoactive combinations (*Figure 7A*), the amount of protein in concentrations where Pikp-2 supresses constitutive cell death may seem lowered.

Nevertheless, Co-IP results depicted a preference in association of Pikp-1 to Pikp-2 over Pikm-2. Increasing concentrations of Pikp-2 reduced the association of Pikp-1 to Pikm-2, outcompeting Pikm-2 from a heterocomplex with Pikp-1 (*Figure 9C*). This correlates with the reduction of the constitutive cell death assay observed in the NLR competition experiments (*Figure 9A and B*).

Altogether, these data reveal that coevolved Pik NLRs display preference in association over non-coevolved NLRs. This represents another example of NLR pair co-adaptation. These differences may underpin the observed cell death phenotypes in response to effectors and in autoimmunity.

## Pik helper/sensor association preference is underpinned by Pik-2 polymorphism

To shed light on the basis of the preferential binding between Pikp-1 and Pikp-2, we tested the role of the polymorphism 230 in this phenotype. For this, we repeated the NLR competition assay co-infiltrating a fixed concentration (OD$_{600}$ 0.4) of Pikp-1 and the autoactive mutant Pikp-2 Asp230Glu, with increasing amounts of Pikp-2.

The combination of Pikp-1 and Pikp-2 Asp230Glu led to a strong cell death in the absence of effector (*Figure 5*). However, increasing concentrations of Pikp-2 significantly reduced this phenotype (*Figure 10*, *Figure 10—source data 1*).

This indicates that the Pik-2 Glu230 polymorphism may also be related to the preferential association between sensor and helper NLRs in addition to its role in specialization towards AVR-Pik effector response and autoimmunity.

## Preferential association in the Pik pair requires the Pik-2 NLR to have a functional P-loop and MHD motifs

To investigate if the preferential sensor/helper association is related to the activation of the helper NLR Pik-2, we tested whether the constitutive cell death mediated by Pikp-2 Asp230Glu could be supressed by mutants that render Pikp-2 inactive.

Although Pik NLRs do not require a functional P-loop or MHD motif to form heterocomplexes (*Figure 8*), we did not observe reduction of cell death phenotypes with increasing concentrations of Pikp-2 mutants in the P-loop (Lys217Arg) or MHD (Asp559Val) motifs (*Figure 11*, *Figure 11—source data 1*, *Figure 11—source data 2*), even at the highest concentration. This indicates that the P-loop and MHD motifs are important for the preferential sensor/helper association observed in Pikp-1 and Pikp-2.

Altogether, these results suggest that changes in Pik-2 helper NLR association to sensor Pik-1 from a resting state into an activated complex require functional P-loop and MHD motifs. This is consistent with studies in the *Arabidopsis* NLR RPP7, where a P-loop mutant retains the ability to associate with autoactive forms of its incompatibility partner HR4 but is not capable of forming higher order assemblies (*Li et al., 2020*).

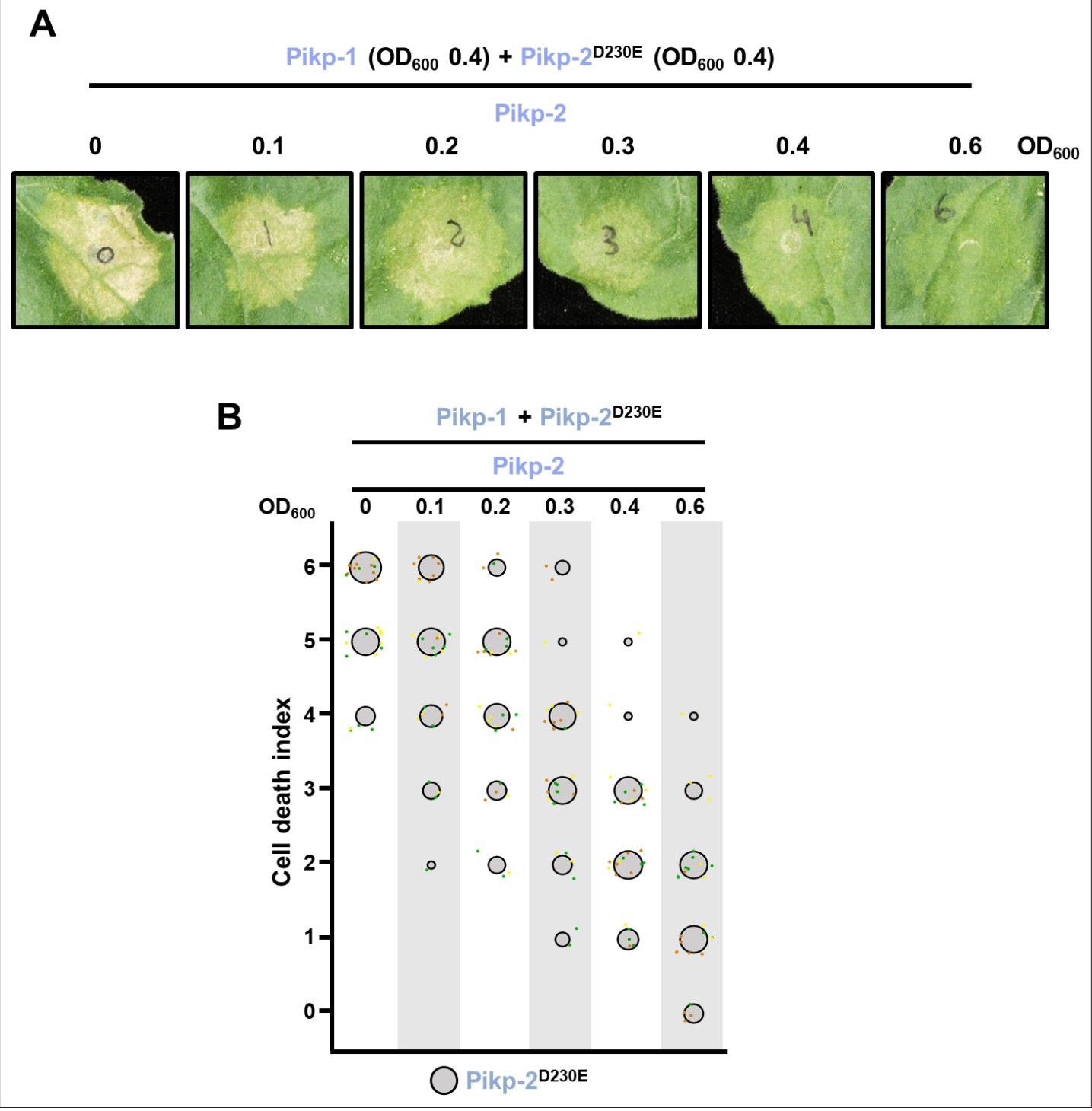

**Figure 10.** Wild-type Pikp-2 supresses constitutive cell death mediated by Pikp-2 Asp230Glu mutant. (**A**) Representative leaf spot images depicting Pikp-2 Asp230Glu-mediated cell death in the presence of Pikp-1 and increasing concentration of Pikp-2. For each experiment, Pikp-1 and Pikp-2 Asp230Glu were co-infiltrated at $OD_{600}$ 0.4 each. Increasing concentrations of Pikp-2 were added to each experiment (from left to right: $OD_{600}$ 0, 0.1, 0.2, 0.3, 0.4, and 0.6). (**B**) Scoring of the cell death mediated by Pikp-2 Asp230Glu in the presence of Pikp-1 and increasing concentration of Pikp-2 assay represented as dot plots. For each experiment, Pikp-1 and Pikp-2 Asp230Glu were co-infiltrated at $OD_{600}$ 0.4 each. Increased concentration of Pikp-2 was added to each experiment (from left to right: $OD_{600}$ 0, 0.1, 0.2, 0.3, 0.4, and 0.6). A total of three biological replicates with 10 internal repeats each were performed for each experiment. For each sample, all the data points are represented as dots with a distinct colour for each of the three biological replicates; these dots are jittered about the cell death score for visualization purposes. The size of the central dot at each cell death value is proportional to the number of replicates of the sample with that score.

*Figure 10 continued on next page*

*Figure 10 continued*

The online version of this article includes the following figure supplement(s) for figure 10:

**Source data 1.** HR scores used for dot plots.

## Discussion

The work presented here highlights sensor/helper coevolution in an allelic rice NLR pair and the basis of their functional diversification towards differential effector recognition specificities (*Figure 12*). We discovered that a single amino acid polymorphism underpins specialization of the helper Pik-2 NLR to its corresponding Pik-1 sensor NLR. Changes in this residue affect cell death outcomes in effector recognition and autoimmune phenotypes. By narrowing down the contribution of NLR specialization to a single amino acid, we could trace the evolutionary history of this polymorphism (*Figure 12*).

The notion that NLRs can work together in pairs is now well established in the field of plant–microbe interactions (*Adachi et al., 2019b*; *Jubic et al., 2019*). Under this emerging framework, it is predicted that cooperating NLRs co-adapt to optimize and maintain a tight control over immune responses. However, the extent to which paired NLRs coevolve to efficiently respond to pathogen effectors while keeping a fine-tuned regulation of immune responses is not well understood at the molecular level. Particularly intriguing is how rapid changes driven by coevolution with pathogen effectors and major evolutionary events, such as the integration of an unconventional domain, impact NLR co-adaptation.

The genetic linkage of the Pik NLR pair has been maintained in grass genomes for tens of millions of years and emerged before the integration of the HMA domain in Pik-1 (*Białas et al., 2021*). This suggests that Pik-1 and Pik-2 have been coevolving for a long time, potentially before providing resistance to the blast fungus. Thus, the integration of the HMA domain in Pik-1, and its subsequent rapid coevolution with rice blast effectors, may have represented a major perturbation on the coevolutionary equilibrium in the paired Pik NLRs. Here, we demonstrated how allelic Pik NLR pairs have differentially coevolved and functionally specialized, leading to autoimmune phenotypes when mismatched. This suggests that in response to HMA integration and diversification in the sensor NLR Pik-1, its helper NLR Pik-2 has acquired polymorphisms to avoid loss of function and/or triggering autoimmunity.

Here, we used Pikp and Pikm as representative examples of the two clades in which Pik alleles are distributed (*Białas et al., 2021*; *De la Concepcion et al., 2021*). Given the similarity between sensor and the helper NLRs within each clade, we predict the phenotypes reported will extend to other similar mismatches between Pik alleles.

To date, integrated domains have been primarily found in paired NLRs that are located in co-regulatory modules with a shared promoter region (*Cesari et al., 2014*). Therefore, the spatial regulation of NLRs with unconventional domains in pairs might be a general mechanism to mitigate NLR misregulation as a consequence of domain integrations or their accelerated evolutionary rates compared with other NLR domains (*Białas et al., 2021*).

We have used *N. benthamiana* as a heterologous system to investigate cell death and autoimmunity in the Pik NLRs. Cell death responses mediated by Pik in this system have previously been shown to correlate with immune responses in rice (*De la Concepcion et al., 2018*; *De la Concepcion et al., 2021*; *Maqbool et al., 2015*). Additional experiments in rice could further clarify the extent to which Pik mismatching leads to autoimmunity and hybrid necrosis. However, given that Pik alleles localize in the same genomic region in different cultivars, and the tight linkage between sensor and helper NLRs (head-to-head orientation with ~3000 bp shared promoter; *Ashikawa et al., 2008*), obtaining rice plants with mismatched combinations of sensor and helper by conventional breeding would be challenging.

Pik autoimmunity also poses the question of whether mismatching between alleles could impose a reproductive barrier. However, to our knowledge, no rice cultivar with mixed Pik alleles has been reported to date (*Figure 6A*). Different allele pairs are present (probably introgressed) in both Japonica and Indica rice varieties. Again, this may be due to difficulties of mixing sensor and helper NLRs in the context of spatial regulation and the tightly genetic linkage.

We found that mismatched allelic NLR pairs can lead to constitutive cell death. We further narrowed down this phenotype to a single Asp to Glu polymorphism, which is the same polymorphism that underpins an extended cell death response to AVR-Pik effectors. Introducing this Asp230Glu polymorphism

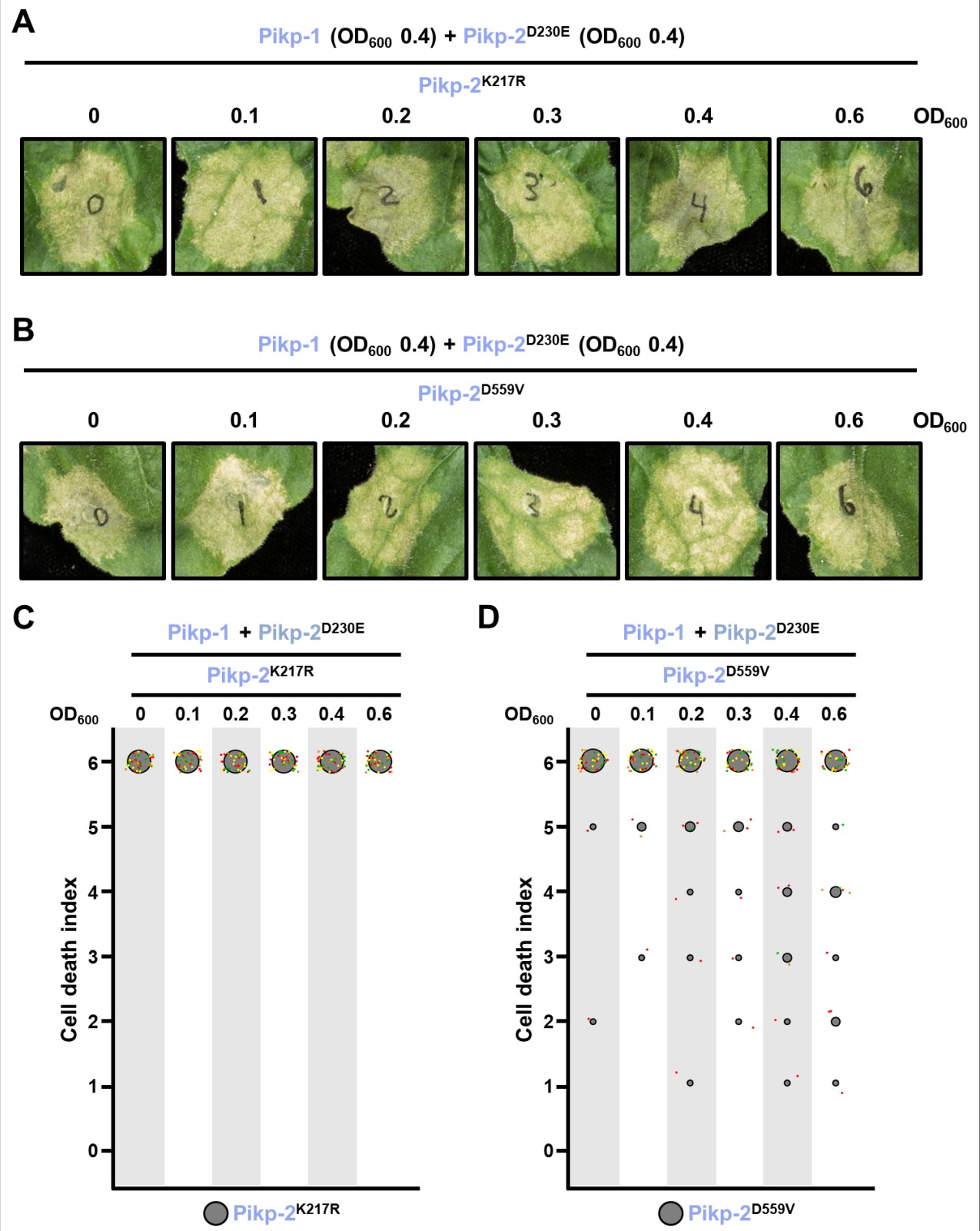

**Figure 11.** Suppression of constitutive cell death mediated by Pikp-2 Asp230Glu requires an active Pikp-2. Representative leaf spot images depicting Pikp-2 Asp230Glu-mediated cell death in the presence of Pikp-1 and increasing concentration of Pikp-2. For each experiment, Pikp-1 and Pikp-2 Asp230Glu were co-infiltrated at $OD_{600}$ 0.4 each. Increasing concentrations of (**A**) Pikp-2 Lys217Arg or (**B**) Pikp-2 Asp559Val were added to each experiment (from left to right: $OD_{600}$ 0, 0.1, 0.2, 0.3, 0.4, and 0.6). Scoring of the cell death mediated by Pikp-2 Asp230Glu in the presence of Pikp-1 and

*Figure 11 continued on next page*

*Figure 11 continued*

increasing concentration of (**C**) Pikp-2 Lys217Arg or (**D**) Pikp-2 Asp559Val represented as dot plots. For each experiment, Pikp-1 and Pikp-2 Asp230Glu were co-infiltrated at $OD_{600}$ 0.4 each. Increased concentration of Pikp-2 mutants was added to each experiment (from left to right: $OD_{600}$ 0, 0.1, 0.2, 0.3, 0.4, and 0.6). A total of four biological replicates with 10 internal repeats each were performed for each experiment. For each sample, all the data points are represented as dots with a distinct colour for each of the four biological replicates; these dots are jittered about the cell death score for visualization purposes. The size of the central dot at each cell death value is proportional to the number of replicates of the sample with that score.

The online version of this article includes the following figure supplement(s) for figure 11:

**Source data 1.** HR scores used for dot plots (*Figure 11C*).

**Source data 2.** HR scores used for dot plots (*Figure 11D*).

in Pikp-2 led to an increase of cell death in response to AVR-Pik effectors as well as to autoimmune phenotypes. As these amino acids have very similar properties, it is intriguing how a fairly minor difference can underpin such a major phenotype. The mechanistic basis of this autoactivation phenotype remains obscure, but it is possible that the larger amino acid side chain (Glu carries an extra methylene group in the side chain) is sufficient to perturb protein–protein interactions that support transition to the active state of the NLR pair. Analogous Asp to Glu changes have been previously shown to act as a gain-of-function mutation in response regulators and transcription factors (*Sakai et al., 2001*; *To et al., 2007*). In some cases, the Asp residue is a target of phosphorylation and the change to Glu partially acts as a phosphomimetic mutation that leads to an active form (*Klose et al., 1993*). Indeed, phosphorylation plays an important role in the activation of the mammalian NLR NLRP3 and the phosphorylation sites are buried in the structure of the resting state (*Hochheiser et al., 2021*). However, to date, there is no evidence to suggest that phosphorylation of Asp230 is involved in Pikp-2 activation. The Asp to Glu change did not prevent sensor/helper association, although it affected association preference. Altogether, this illustrates that small changes in NLR receptors can have profound phenotypical effects on immune regulation and cell death responses.

We still lack detailed information about the activation mechanism of paired plant NLRs although a cooperation mechanism has been proposed for the Pik NLR pair (*Zdrzałek et al., 2020*). By taking advantage of constitutively active Pik sensor/helper combinations and mutants, we can expand our knowledge of NLR signalling mechanisms. The use of constitutively active immune receptors as a research tool is starting to be explored in the field of NLR biology. This approach has the advantage of simplifying the complex requirements of immune activation by removing the variability of the effector.

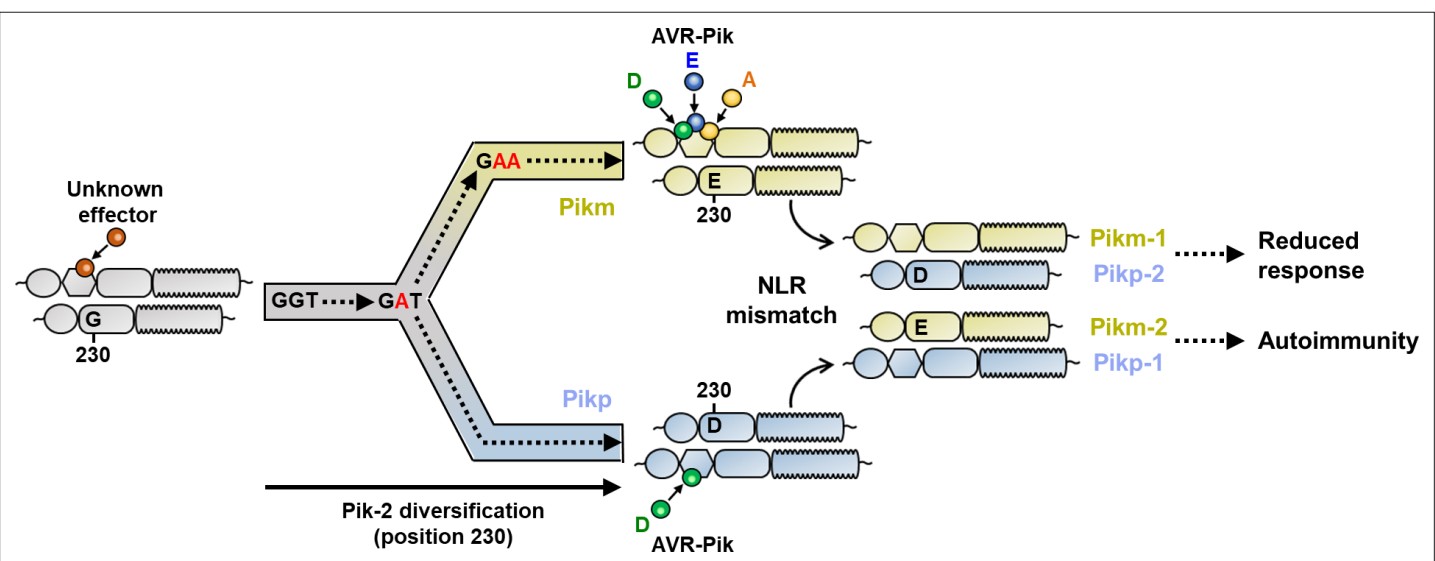

**Figure 12.** Schematic representation of the proposed evolutionary model of the Pik pairing. Pikp (coloured in ice blue) and Pikm (coloured in gold) have evolved and specialized from an ancestral NLR pair (coloured in grey), functionally diversifying and gaining recognition to a different subset of allelic AVR-Pik effectors. Residues at Pik-2 polymorphic position 230 are indicated, and mutations predicted to have occurred during this transition are indicated in red. As a consequence of diversification, mismatch of Pikp and Pikm impairs immune responses and leads to NLR autoactivation and constitutive cell death in *N. benthamiana*.

**eLife** Research article

Plant Biology

It also renders full receptor activation, whilst relying solely on effector recognition can provide a mixture of active and inactive receptors.

Sensor and helper Pik NLRs form a pre-activation complex (*Zdrzałek et al., 2020*). Activation of immune responses may rearrange the composition of this complex, possibly affecting sensor/helper stoichiometry, as described for NAIP/NLRC4 inflammasomes (*Hu et al., 2015*; *Tenthorey et al., 2017*; *Zhang et al., 2015*). This rearrangement is dependent on nucleotide binding and has been fine-tuned during the evolutionary process, as depicted in the competition assays in the Pik pair.

Autoactive mutations in Pikp-2 led us to re-evaluate the involvement of conserved P-loop and MHD motif of sensor and helper Pik NLRs in signalling activation. In contrast to the previously described cooperation mechanism of Pik regulation (*Białas et al., 2018*; *Zdrzałek et al., 2020*), the P-loop of sensor NLR Pik-1 is important but not necessary for NLR activation. This also deviates from the negative regulation mechanism described for other NLR pairs such as Pia or *Arabidopsis* RRS1/RPS4 (*Césari et al., 2014*; *Cesari et al., 2013*; *Le Roux et al., 2015*; *Sarris et al., 2015*) and suggests that the Pik pair may trigger cell death via a different mechanism. Surprisingly, combining Pik-2 mutants in the P-loop and MHD motifs within the autoactive Pikp-1/Pikp-2 Asp230Glu background did not lead to a reduction in constitutive cell death, as observed with the Pikp-2 wild-type (*Figure 11*). This suggests that these mutants cannot outcompete Pikp-2 Asp230Glu from a complex with Pikp-1, meaning that mutations in these domains, although they do not prevent association, may affect the strength of sensor/helper association. However, the requirements of these regions for the formation of a hypothetical NLR complex remain unclear. Future experiments using autoactive combinations should help unravel the requirements of Pik NLR immune signalling and will reveal the nature of the changes undergone by NLRs during activation.

While Pikp-2 Asp230Glu induces a strong constitutive cell death when paired with Pikp-1, it only leads to a weak cell death response in the absence of the effector when paired with Pikm-1. This may reflect another layer of sensor/helper coevolution where sensor Pikm-1 may have adapted to supress uncontrolled activation triggered by the Asp230Glu polymorphism found in Pikm-2. Given the similarity between Pikp-1 and Pikm-1 outside the integrated HMA (*Białas et al., 2021*), this domain may harbour the compatibility determinant with this polymorphism. Integrated domains may have regulatory functions other than binding effectors (*Ma et al., 2018*). Further comparative analysis between Pikp and Pikm HMA domains, using the repertoire of mutants developed here, will be required to address this question that is important for future engineering efforts based on the Pik system.

In summary, this work provides an evolutionary framework for how differential selective pressures, such as recognition of pathogen strains via effector binding, impact NLRs pairs. It uncovers the potential of paired NLRs to give rise to autoimmune phenotypes during evolution and links pathogen perception and autoimmunity.

# Materials and methods

## Key resources table

| Reagent type (species) or resource | Designation | Source or reference | Identifiers | Additional information |
|---|---|---|---|---|
| Recombinant DNA reagent | pICH47742 | Addgene | | |
| Recombinant DNA reagent | pICH47751 | Addgene | | |
| Commercial assay or kit | pCR8/GW/TOPO TA Cloning Kit | Thermo Fisher | K250020 | |
| Commercial assay or kit | ANTI-FLAG M2 Affinity Agarose Gel | Sigma (Merck) | A2220 | |
| Antibody | ANTI-FLAG M2 antibody (mouse monoclonal) | Sigma (Merck) | Cat. # F1804; lot # SLBT7654 | Used diluted (1:3000) |
| Antibody | Anti-HA high-affinity antibody 3F10 (rat monoclonal) | Roche | Cat. # 11867423001; lot # 14553800 | Used diluted (1:3000) |

*Continued on next page*

*Continued*

| Reagent type (species) or resource | Designation | Source or reference | Identifiers | Additional information |
|---|---|---|---|---|
| Antibody | V5 Tag Antibody (E10/V4RR), HRP conjugated (mouse monoclonal) | Invitrogen | MA5-15253-HRP | Used diluted (1:3000) |
| Antibody | Anti-Rat IgG-Peroxidase antibody produced in goat | Sigma (Merck) | Cat. # A9307 | Used diluted (1:10,000) |
| Antibody | Anti-mouse IgG, HRP conjugate | Promega | Cat. # W4021 | Used diluted (1:10,000) |
| Commercial assay or kit | ECL extreme Lumiblue Western Blotting Substrate | Abcam | Ab270517 | |
| Software, algorithm | besthr | *De la Concepcion et al., 2019* | | |
| Software, algorithm | iTOL v5.5.1 | *Letunic and Bork, 2019* | | |
| Software, algorithm | RAxML v8.2.11 | *Stamatakis, 2014* | | |
| Software, algorithm | *ggplot2* R package | | | |
| Software, algorithm | QKphylogeny | https://github.com/matthewmoscou/QKphylogeny | | |

## Phylogenetics analyses

Codon-based alignment was generated using MUSCLE 3.8.425 (*Edgar, 2004*). The alignment positions with more than 40% data missing were removed using QKphylogeny (https://github.com/matthewmoscou/QKphylogeny; *Moscou, 2021* copy archived at swh:1:rev:36ad9a44c761f046e413b-60b21e4ce5452f2bb11). The maximum likelihood tree was calculated from a 3066-nt-long alignment using 1000 bootstrap method (*Felsenstein, 1985*) and GTRGAMMA substitution model (*Tavaré, 1986*) as implemented in RAxML v8.2.11 (*Stamatakis, 2014*). Best-scoring tree was manually rooted using the Pik-2 sequence from *Leersia perreri* and visualized using the iToL tool v5.5.1 (*Letunic and Bork, 2019*). The interactive tree is publicly available at https://itol.embl.de/tree/8229133147185181 615486010.

Joint reconstruction of ancestral sequences (*Yang et al., 1995*), based on the algorithm of *Pupko et al., 2000*, was performed using the codeml program as part of the PAML 4.9j package (*Yang, 1997*). The ancestral sequence reconstruction was carried out based on best-scoring ML tree and a 3261-nt-long codon alignment of the full-length Pik-2 sequences.

The accession numbers of the sequences used in the phylogenetic analyses are LPERR11G19580.2, ONIVA11G22700, ORUFI11G24740, XM_015762499.2, OGLUM11G22330, ORGLA11G0185700, MW568036, MW568041, MW568042, MW568043, MW568044, MW568045, KN541092.1, OPUNC11G19560, OBART11G23160.1, GU811862, HQ606329, HM048900_1, HQ662329_1, GU811867, AB462325, GU811861, GU811864, GU811865, GU811866, HQ662330, HM035360, KU365338.1, HQ660231, GU811868, GU811869, GU811870, GU811871, and GU811872.

## Gene cloning

For protein expression in planta, we used full-length Pikp-1 and Pikm-1 into the plasmid pICH47742 with a C-terminal 6×His/3×FLAG tag as previously described (*De la Concepcion et al., 2018*). Wild-type Pikp-2 and Pikm-2 in pICH47751 with C-terminal 6×HA were also described in *De la Concepcion et al., 2018*, and Pik-2 mutated versions were generated by site-directed mutagenesis (see below) using appropriate Pik-2 template in pCR8/GW/TOPO (Invitrogen) with Golden Gate compatible overhangs. The constructs were later assembled in pICH47751 under control of *A. tumefaciens* mannopine

synthase (Mas) promoter and terminator and a C-terminal 6×HA using golden gate cloning (*Engler et al., 2014*). AVR-Pik effector alleles used in this study were previously described in *De la Concepcion et al., 2018*.

All DNA constructs were verified by sequencing.

## Site-directed mutagenesis

Point mutations were introduced in Pik-2 by PCR amplification with Phusion polymerase (Thermo Fisher Scientific) using 5'-phosphorylated primers carrying the desired mutations. The amplification used primers running in opposite directions from the mutation site in the template Pik-2 in pCR8/GW/TOPO vector (Invitrogen). DNA templates were then eliminated by incubating the reaction with DpnI (New England Biolabs) for 1 hr at 37°C. After PCR purification of the amplified products, the DNA sequence was re-ligated using T4 DNA ligase (New England Biolabs) according to the manufacturer's protocol in 20 µl reactions incubated overnight at room temperature. Competent *Escherichia coli* DH5α cells were subsequently transformed with 5 µl of the reaction. The resulting constructs were sequenced to ensure that a correct mutation was inserted into the sequence.

## In planta Co-IP

Transient gene expression in planta for Co-IP was performed by delivering T-DNA constructs with *A. tumefaciens* GV3101 (C58 (rifR) Ti pMP90 (pTiC58DT-DNA) (gentR) Nopaline (pSoup-tetR)) strain into 4-week-old *N. benthamiana* plants grown at 22–25°C with high light intensity. *A. tumefaciens* strains carrying the given wild-type or mutated Pik-1 or Pik-2 were infiltrated at $OD_{600}$ 0.2 each (unless otherwise stated), in agroinfiltration medium (10 mM $MgCl_2$, 10 mM 2-(N-morpholine)-ethanesulfonic acid [MES], pH 5.6), supplemented with 150 µM acetosyringone.

For detection of complexes in planta, leaf tissue was collected 2–3 days post infiltration (dpi), frozen, and ground to fine powder in liquid nitrogen using a pestle and mortar. Leaf powder was mixed with two times weight/volume ice-cold extraction buffer (10% glycerol, 25 mM Tris pH 7.5, 1 mM EDTA, 150 mM NaCl, 2% w/v PVPP, 10 mM DTT, 1× protease inhibitor cocktail [Sigma], 0.1% Tween 20 [Sigma]), centrifuged at 4200 × *g* at 4°C for 20–30 min, and the supernatant was passed through a 0.45 µm Minisart syringe filter. The presence of each protein in the input was determined by SDS-PAGE/western blot. Wild-type and mutated Pik-1 and Pik-2 proteins were detected probing the membrane with anti-FLAG M2 antibody (Sigma) and anti-HA high-affinity antibody 3F10 (Roche), respectively. For detection of Pikm-2 in the competition experiments in planta, we used anti-V5 antibody HRP-conjugated (Invitrogen).

For immunoprecipitation, 1.5 ml of filtered plant extract was incubated with 30 µl of M2 anti-FLAG resin (Sigma) in a rotatory mixer at 4°C. After 3 hr, the resin was pelleted (800 g, 1 min) and the supernatant removed. The pellet was washed and resuspended in 1 ml of IP buffer (10% glycerol, 25 mM Tris pH 7.5, 1 mM EDTA, 150 mM NaCl, 0.1% Tween 20 [Sigma]) and pelleted again by centrifugation as before. Washing steps were repeated five times. Finally, 30 µl of LDS Runblue sample buffer was added to the agarose and incubated for 10 min at 70°C. The resin was pelleted again, and the supernatant loaded on SDS-PAGE gels prior to western blotting. Membranes were probed with anti-FLAG M2 antibody (Sigma) and anti-HA high-affinity antibody 3F10 (Roche) monoclonal antibodies. For competition experiments, the membrane was additionally probed with anti-V5 antibody HRP-conjugated (Invitrogen) to detect Pikm-2.

## *N. benthamiana* cell death assays

*A. tumefaciens* GV3101 (C58 [rifR] Ti pMP90 [pTiC58DT-DNA] [gentR] Nopaline [pSoup-tetR]) carrying wild-type or mutated Pik-1 and Pik-2 were resuspended in agroinfiltration media (10 mM $MgCl_2$, 10 mM MES, pH 5.6) supplemented with 150 µM acetosyringone. Given combinations of Pik-1 and Pik-2 constructs were mixed at $OD_{600}$ 0.4 for each construct. *A. tumefaciens* GV3101 carrying AVR-Pik effectors or mCherry were added to each experiment at $OD_{600}$ 0.6. Each infiltration had additional *A. tumefaciens* GV3101 (C58 [rifR] Ti pMP90 [pTiC58DT-DNA] [gentR] Nopaline [pSoup-tetR]) carrying P19 at $OD_{600}$ 0.1. Leaves of 4-week-old *N. benthamiana* were infiltrated using a needleless syringe. Leaves were collected at 5 dpi to measure UV autofluorescence as proxy for cell death as reported previously (*De la Concepcion et al., 2019*; *De la Concepcion et al., 2018*; *Maqbool et al., 2015*).

## Cell death scoring: UV autofluorescence

Detached leaves were imaged at 5 dpi from the abaxial side of the leaves for UV fluorescence images. Photos were taken using a Nikon D4 camera with a 60 mm macro lens, ISO set 1600 and exposure ~10 s at F14. The filter is a Kodak Wratten No. 8 and white balance is set to 6250 degrees Kelvin. Blak-Ray longwave (365 nm) B-100AP spotlight lamps are moved around the subject during the exposure to give an even illumination. Images shown are representative of three independent experiments, with internal repeats. The cell death index used for scoring is as presented previously (*Maqbool et al., 2015*). Dot plots were generated using R v3.4.3 (https://www.r-project.org/) and the graphic package ggplot2 (*Wickham, 2016*). The size of the centre dot at each cell death value is directly proportional to the number of replicates in the sample with that score. All individual data points are represented as dots.

## Statistical analyses

Cell death scoring from autofluorescence was analysed using estimation methods (*Ho et al., 2019*) and plotted using the besthr R library as implemented before (*De la Concepcion et al., 2019*). All cell death scores in samples under comparison were ranked, irrespective of sample. The mean ranks of the control and test sample were taken and a bootstrap process was begun on ranked test data, in which samples of equal size to the experiment were replaced and the mean rank was calculated. After 1000 bootstrap samples, rank means were calculated, a distribution of the mean ranks was drawn, and its 2.5 and 97.5 quantiles calculated. If the mean of the control data is outside of these boundaries, the control and test means were considered to be different.

## Acknowledgements

We thank present and former members of the Banfield and Kamoun laboratories for discussions that have shaped this manuscript, and colleagues at Iwate Biotechnology Research Center for stimulating discussions on NLR biology. We specially thank Dr. Cristina Barragan and Dr. Adam Bentham for critical reading of the manuscript. We also thank Andrew Davies and Phil Robinson from JIC Scientific Photography for the UV pictures of the cell death assays. This work was supported by the Biotechnology and Biological Sciences Research Council (BBSRC, UK, grant BB/012574, BBS/E/J/000PR9795), the BBSRC Doctoral Training Partnership at Norwich Research Park (grant: BB/M011216/1, project reference 1771322), the European Research Council (proposal 743165), the John Innes Foundation, the Gatsby Charitable Foundation, the European Commission through the Erasmus+ programme, and JSPS Grant 20H05681.

## Additional information

### Competing interests

Sophien Kamoun: SK receives funding from industry on NLR biology. The other authors declare that no competing interests exist.

### Funding

| Funder | Grant reference number | Author |
| --- | --- | --- |
| Biotechnology and Biological Sciences Research Council | BB/012574 | Sophien Kamoun Mark J Banfield |
| Biotechnology and Biological Sciences Research Council | BBS/E/J/000PR9795 | Sophien Kamoun Mark J Banfield |
| Biotechnology and Biological Sciences Research Council | BB/M011216/1 | Aleksandra Bialas |

| Funder | Grant reference number | Author |
|---|---|---|
| ERC Horizon 2020 | 743165 | Sophien Kamoun<br>Mark J Banfield |
| John Innes Foundation | | Juan Carlos De la Concepcion |
| Gatsby Charitable Foundation | | Sophien Kamoun |
| Erasmus+ | | Javier Vega Benjumea |
| Japan Society for the Promotion of Science | 20H05681 | Ryohei Terauchi |
| Biotechnology and Biological Sciences Research Council | project reference 1771322 | Aleksandra Bialas |

The funders had no role in study design, data collection and interpretation, or the decision to submit the work for publication.

### Author contributions
Juan Carlos De la Concepcion, Conceptualization, Data curation, Formal analysis, Investigation, Methodology, Project administration, Supervision, Validation, Visualization, Writing – original draft, Writing – review and editing; Javier Vega Benjumea, Funding acquisition, Investigation, Methodology, Writing – review and editing; Aleksandra Bialas, Formal analysis, Investigation, Writing – review and editing; Ryohei Terauchi, Funding acquisition, Writing – review and editing; Sophien Kamoun, Conceptualization, Resources, Supervision, Writing – review and editing; Mark J Banfield, Conceptualization, Funding acquisition, Methodology, Project administration, Resources, Supervision, Validation, Writing – original draft, Writing – review and editing

### Author ORCIDs
Juan Carlos De la Concepcion http://orcid.org/0000-0002-7642-8375
Sophien Kamoun http://orcid.org/0000-0002-0290-0315
Mark J Banfield http://orcid.org/0000-0001-8921-3835

### Decision letter and Author response
Decision letter https://doi.org/10.7554/eLife.71662.sa1
Author response https://doi.org/10.7554/eLife.71662.sa2

---

# Additional files

### Supplementary files
• Transparent reporting form
• Source data 1. Raw uncropped images and uncropped labelled images for all western blots.

### Data availability
All data generated or analysed during this study are included in the manuscript and supporting files. Source data files have been provided for Plots and Blots.

---

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
