## [Editor Report]

De la Concepcion and colleagues investigated the mode of co-evolution of plant immune receptor pair that functions as a unit to detect pathogen invasion and turn on immunity. The study shows that an allelic mismatch of a receptor pair from rice can cause autoimmunity in the absence of pathogen effectors, and this can be traced to polymorphisms that arose fairly recently. Overall the study supports that the paired receptors have coevolved to prevent premature inactivation.

---

## [Decision Letter]

**Decision letter after peer review:**

Thank you for submitting your article "Functional diversification gave rise to allelic specialization in a rice NLR immune receptor pair" for consideration by *eLife*. Your article has been reviewed by 3 peer reviewers, including Jian-Min Zhou as Reviewing Editor and Reviewer #1, and the evaluation has been overseen by Jürgen Kleine-Vehn as the Senior Editor. The following individual involved in review of your submission has agreed to reveal their identity: Eunyoung Chae (Reviewer #2).

Essential revisions:

1) The autoimmunity was observed solely in the heterologous system and this may or may not reflect hybrid necrosis in rice. Does the paired NLR incompatibility play a role in setting a reproductive barrier between japonica and indica of modern rice varieties? Such data will significantly strengthen the paper. If not, it is helpful to discuss.

2) On the mechanistic side, the co-IP experiments fall short to show how the mismatch at position 230 of Pik-2 leads to autoimmunity. Do they form hetero-oligomers in the absence of effectors? This experiment could be performed in a short period of time.

*Reviewer #1 (Recommendations for the authors):*

Please see public review.

*Reviewer #2 (Recommendations for the authors):*

Data presentation was carefully made to deliver authors' claims with robust quantification of HR readouts. The manuscript was well written so that hypotheses being tested could be appreciated with nice graphics. I would like to provide several comments on which authors may consider revising the manuscript to improve some parts of it.

1. Out of curiosity, I would like to ask if the paired NLR incompatibility might have played a role in setting a reproductive barrier between japonica and indica of modern rice varieties. This notion can be included in the discussion if the authors have considered to look into a reproductive barrier between Pikp-1 carriers and Pikm-2 carriers. At least, indicating japonica vs. indica labels in Figure 6A Figure 6 (blue light and golden clades) would help so that the readers could associate the presence (or fixture) of the respective allelic pairs with japonica/indica varieties.

2. Observing suppression of immunity can be equally plausible to that of autoimmunity when it comes to a mismatched NLR pair. Although "reduced responses" was conceptually depicted in Figure 12, it has not been elaborated in results. Is there a chance that the authors had observed a broad scale of suppression of immunity when AvrPik (other than D) was combined with any mismatched Pik allelic combination? I do not think it is necessary to include a full set of data but the notion of Figure 12 can be clarified with some textual elaboration by linking to the presented data. Alternatively, the statement in L213-214 can be described with details.

3. I would recommend to revise the sections regarding Figures 10 and 11, preferably combining Figure 10 and Figure 11. Figure 11 alone and its conclusion as a separate section can be confusing if not compared with results in Figure 10.

a. Regarding the section under the title of "Pik pair preferential association requires NLR activation" (L428-433)

The provided data in Figure 11 suggests that the autoactive Pikp-1/Pikp-2 D230E complex cannot be titrated out with the presence of Pikp-2 mutants of P-loop or of MHV (supposedly defective in NLR activation and thus signaling), as opposed to WT Pikp-2 (230 GLU) as a competitor (Figure 10). These data set (Figures 10 and 11) reinforces the preferential association with "co-evolutionarily" elaborated partner NLR while the preference requires a full functional NLR. The data can be simply interpreted as the mutants, despite being 230 GLU, just not being able to compete with Pikp-2 D230E. The authors may be able to "discuss" the results in IP/biochemical data here to clarify the association strength difference between Pikp-1 and Pikp-2 D230E vs. between Pikp-1 and Pikp-2 P-loop/MHV mutants to exclude merely the possibility of non-titration. From the citation of RPW8/RPP7 work, I got to know that authors assumed that Pikp-1+ Pikp-2 P-loop/MHV mutant (non-functional) was able to titrate out the D230E and form a complex. But in this case, the HR signal should not be there, as Pikp-2 P-loop as compared to Pikp-1 seems to be relatively more important ones for signaling.

b. It is still not clear if the preferential association "require NLR activation" or merely require NLRs that have a full potential to activate. The current subtitle is misleading in my opinion as it puts a premise that an NLR needs to be activated to be preferentially associated with the partner. As the authors indicated that pre-active NLR complex can form as an oligomeric status, "NLR activation" as a complex can occur even after pre-assembly. The authors are advised to revise the subsection title, since "NLR activation" can mean either conformation change of an single NLR molecule to oligomerize or preassemble NLR complex to be activated.

c. L432-433 Do authors refer to P-loop and MHD motif in both Pikp-1 /Pikp-2 or either of them? An alternative explanation of HR not being titrated out with non-functional Pik-2 in Figure 11 can be due to Pikp-1 still providing functional P-loop, despite not absolutely necessary?

*Reviewer #3 (Recommendations for the authors):*

As far as I understood, the submitted manuscript does not examine how the polymorphic residue in the helper Pik regulates the perception of AVR by sensor Pik and presumptive oligomerization of helper Pik after AVR perception. I assume that "in planta" interaction between sensor Pik and AVR in the presence of helper Pik allelic variant could be tested by Co-IP (or other means). Also examining the presumed "in planta" oligomerisation (i.e., self-association) of helper Pik allelic variants after AVR perception will be helpful to understand the polymorphic residue. I believe that these data would serve as a template for other NLR pair studies.

---

## [Author Response]

Essential revisions:1) The autoimmunity was observed solely in the heterologous system and this may or may not reflect hybrid necrosis in rice. Does the paired NLR incompatibility play a role in setting a reproductive barrier between japonica and indica of modern rice varieties? Such data will significantly strengthen the paper. If not, it is helpful to discuss.

We acknowledge that our functional assays for immune activation have been performed in the heterologous system *N. benthamiana* and that extending these experiments to rice would be ideal. However, it is important to note that the Pik pair of genes are in a head-to-head orientation separated only by their shared promotor (~3000 bp). Thus, such experiments are far from trivial and would involve a lengthy process that still might not yield a clear outcome given the characteristics of the Pik locus. Additionally, the cell death outcome in the *N. benthamiana* system has been shown to correlate well with Pik-mediated immune activation in rice (De la Concepcion et al., 2018; De la Concepcion et al., 2021), providing a powerful platform for us to use high-throughput experiments involving multiple mutants and several sensor/helper combinations.

Whether autoimmunity between alleles has the potential to establish a reproductive barrier in modern rice is unknown. So far, no cultivars with mixed Pik alleles have been reported, but this may be because these alleles localize in the same genomic region and sensor/helper pairs are in tight association, making it difficult for them to be mixed by common breeding. However, Pik alleles are present (probably introgressed) in both Japonica and Indica cultivars (we added this extra information in Figure 6). Therefore we can exclude the possibility that Pik establishes a reproductive barrier between them.

In response to this essential revision, we have expanded our discussion to reflect the limitation of using a heterologous system and added information concerning the possibility of Pik acting as a Japonica/Indica reproductive barrier.

2) On the mechanistic side, the co-IP experiments fall short to show how the mismatch at position 230 of Pik-2 leads to autoimmunity. Do they form hetero-oligomers in the absence of effectors? This experiment could be performed in a short period of time.

We agree that, as it is, the mechanistic basis of Pik NLR autoimmunity is not fully explained in this manuscript. We had already included an extended discussion on this in the original submission (from 507 to 712 in the revised manuscript) to ensure this is clearly appreciated, but we also speculate about the different possibilities.

Regarding the specific co-IP experiments mentioned, to test whether helper NLR mutants at the position 230, associate (forming hetero-oligomers) with WT sensor NLRs Pikp-1 and Pikm-1 in the absence of effector. Such experiments were shown in the original submission in Figure 7 —figure supplement 1. We found that there were no major changes in association induced by mutations at this position and speculated in the discussion that minor changes (via an unknown mechanism) are responsible for the cell death phenotype.

Elucidation of the mechanistic basis of the Pik autoimmune phenotype (as well as AVR-Pik triggered immunity) requires a deeper understanding of sensor/helper association and most likely the structural changes that occur during activation, which are beyond the scope of the work presented here.

Reviewer #2 (Recommendations for the authors):Data presentation was carefully made to deliver authors' claims with robust quantification of HR readouts. The manuscript was well written so that hypotheses being tested could be appreciated with nice graphics. I would like to provide several comments on which authors may consider revising the manuscript to improve some parts of it.1. Out of curiosity, I would like to ask if the paired NLR incompatibility might have played a role in setting a reproductive barrier between japonica and indica of modern rice varieties. This notion can be included in the discussion if the authors have considered to look into a reproductive barrier between Pikp-1 carriers and Pikm-2 carriers. At least, indicating japonica vs. indica labels in Figure 6A (blue light and golden clades) would help so that the readers could associate the presence (or fixture) of the respective allelic pairs with japonica/indica varieties.

We do not know whether Pik alleles may form a reproductive barrier between Indica and Japonica cultivars. However, it seems unlikely as Pik alleles are present in both Indica and Japonica varieties (probably introgressed by breeding). We have added this information to the labels in Figure 6A as suggested by the reviewer and added comments on this to the discussion.

2. Observing suppression of immunity can be equally plausible to that of autoimmunity when it comes to a mismatched NLR pair. Although "reduced responses" was conceptually depicted in Figure 12, it has not been elaborated in results. Is there a chance that the authors had observed a broad scale of suppression of immunity when AvrPik (other than D) was combined with any mismatched Pik allelic combination? I do not think it is necessary to include a full set of data but the notion of Figure 12 can be clarified with some textual elaboration by linking to the presented data. Alternatively, the statement in L213-214 can be described with details.

We are a little unsure what the reviewer is referring to here. We observed a general reduced response to effectors in the Pikm-1/Pikp-2 mismatch, although we suggest this is due to lower activation of Pikp-2 compared to Pikm-2, supported by the experiments with single point mutants.

3. I would recommend to revise the sections regarding Figures 10 and 11, preferably combining Figure 10 and Figure 11. Figure 11 alone and its conclusion as a separate section can be confusing if not compared with results in Figure 10.

We thank the reviewer for the suggestion. We actually had these figures combined in earlier versions of the manuscript, but we felt it made more sense to separate them as they analyse the effect of different types of mutations (single mutant in the polymorphism 230 versus mutations in conserved p-Loop/MHD domains).

a. Regarding the section under the title of "Pik pair preferential association requires NLR activation" (L428-433)The provided data in Figure 11 suggests that the autoactive Pikp-1/Pikp-2 D230E complex cannot be titrated out with the presence of Pikp-2 mutants of P-loop or of MHV (supposedly defective in NLR activation and thus signaling), as opposed to WT Pikp-2 (230 GLU) as a competitor (Figure 10). These data set (Figures 10 and 11) reinforces the preferential association with "co-evolutionarily" elaborated partner NLR while the preference requires a full functional NLR. The data can be simply interpreted as the mutants, despite being 230 GLU, just not being able to compete with Pikp-2 D230E. The authors may be able to "discuss" the results in IP/biochemical data here to clarify the association strength difference between Pikp-1 and Pikp-2 D230E vs. between Pikp-1 and Pikp-2 P-loop/MHV mutants to exclude merely the possibility of non-titration. From the citation of RPW8/RPP7 work, I got to know that authors assumed that Pikp-1+ Pikp-2 P-loop/MHV mutant (non-functional) was able to titrate out the D230E and form a complex. But in this case, the HR signal should not be there, as Pikp-2 P-loop as compared to Pikp-1 seems to be relatively more important ones for signaling.

We expect a titration as the p-Loop/MHD mutants do associate with the sensor NLR. As we saw no effect of the titration, we concluded that this may indicate these domains may be important for the strength of association. We have commented about this result in the discussion.

b. It is still not clear if the preferential association "require NLR activation" or merely require NLRs that have a full potential to activate. The current subtitle is misleading in my opinion as it puts a premise that an NLR needs to be activated to be preferentially associated with the partner. As the authors indicated that pre-active NLR complex can form as an oligomeric status, "NLR activation" as a complex can occur even after pre-assembly. The authors are advised to revise the subsection title, since "NLR activation" can mean either conformation change of an single NLR molecule to oligomerize or preassemble NLR complex to be activated.

We changed the subtitle accordingly.

c. L432-433 Do authors refer to P-loop and MHD motif in both Pikp-1 /Pikp-2 or either of them? An alternative explanation of HR not being titrated out with non-functional Pik-2 in Figure 11 can be due to Pikp-1 still providing functional P-loop, despite not absolutely necessary?

We referred to Pikp-2 only and have changed the text to clarify. We have not investigated the effect of Pikp-1 P-loop mutations in the titration. It is an interesting alternative explanation; however, we would expect that if Pik-1 was providing a functional P-loop, we would see titration at least for the Pik-2 P-loop mutant.

Reviewer #3 (Recommendations for the authors):As far as I understood, the submitted manuscript does not examine how the polymorphic residue in the helper Pik regulates the perception of AVR by sensor Pik and presumptive oligomerization of helper Pik after AVR perception. I assume that "in planta" interaction between sensor Pik and AVR in the presence of helper Pik allelic variant could be tested by Co-IP (or other means). Also examining the presumed "in planta" oligomerisation (i.e., self-association) of helper Pik allelic variants after AVR perception will be helpful to understand the polymorphic residue. I believe that these data would serve as a template for other NLR pair studies.

We thank the reviewer for taking the time to read our manuscript. We hope we have addressed most of their concerns in response to other reviewers. In particular where we highlight the limitations of our current study, but how the experiments are important to inform future work on the mechanistic basis of how effector perception activates the Pik NLR pair, and the importance of specialisation in sensor-helper pairing. Inferring the association of Pik-2 helpers will be the focus of future experiments.

References

Białas, A., Langner, T., Harant, A., Contreras, M.P., Stevenson, C.E.M., Lawson, D.M., Sklenar, J., Kellner, R., Moscou, M.J., Terauchi, R.*, et al.* (2021). Two NLR immune receptors acquired high-affinity binding to a fungal effector through convergent evolution of their integrated domain. *eLife 10*, e66961.

De la Concepcion, J.C., Franceschetti, M., MacLean, D., Terauchi, R., Kamoun, S., and Banfield, M.J. (2019). Protein engineering expands the effector recognition profile of a rice NLR immune receptor. e*Life 8*.

De la Concepcion, J.C., Franceschetti, M., Maqbool, A., Saitoh, H., Terauchi, R., Kamoun, S., and Banfield, M.J. (2018). Polymorphic residues in rice NLRs expand binding and response to effectors of the blast pathogen. Nat Plants *4*, 576-585.

De la Concepcion, J.C., Maidment, J.H.R., Longya, A., Xiao, G., Franceschetti, M., and Banfield, M.J. (2021). The allelic rice immune receptor Pikh confers extended resistance to strains of the blast fungus through a single polymorphism in the effector binding interface. PLOS Pathogens *17*, e1009368.

Zdrzalek, R., Kamoun, S., Terauchi, R., Saitoh, H., and Banfield, M.J. (2020). The rice NLR pair Pikp-1/Pikp-2 initiates cell death through receptor cooperation rather than negative regulation. PloS one *15*.